# Advancing snow data assimilation with a dynamic observation uncertainty

Devon Dunmire[1], Michel Bechtold[1], Lucas Boeykens[1,2], and Gabriëlle J. M. De Lannoy[1]

[1]Department of Earth and Environmental Sciences, KU Leuven, Leuven, Belgium
[2]Department of Environment, Ghent University, Ghent, Belgium

**Correspondence:** Devon Dunmire (devon.dunmire@kuleuven.be)

**Abstract.** Seasonal snow is a critical resource for society by providing water for billions, supporting agriculture, clean energy, and tourism, and is an important element within the climate system by influencing the global energy balance. However, accurately quantifying snow mass, particularly in mountainous regions, remains a challenge due to substantial observational and modeling limitations. As such, data assimilation (DA) offers a powerful solution by integrating observations with physically-based models to improve estimates of the snowpack. Previous snow DA studies have employed an Ensemble Kalman Filter (EnKF) to assimilate Sentinel-1 satellite-based snow depth retrievals, demonstrating improved accuracy in modeled snow depth, mass, and streamflow when evaluated against in-situ measurements. In those studies, the uncertainty of the assimilated retrievals was assumed to be static in time and space, likely leading to a suboptimal use of the observational information. Here, we present several advances in snow DA. Using an EnKF, we assimilate novel snow depth retrievals derived from a machine learning product that leverages Sentinel-1 backscatter observations, land cover, and topographic information over the European Alps. We also incorporate a spatiotemporally dynamic observation error, whereby the uncertainty of the assimilated snow depth retrieval varies in space and time with snow depth ($DA_{var}$ experiment). The machine learning snow depth retrieval product is assimilated into the Noah-MP land surface model over the entire European Alps at 1 km resolution for the years 2015-2023 and snow depth, snow water equivalent, and snow cover are evaluated against independent in-situ data and satellite observations. The $DA_{var}$ experiment offers small, but significant improvements to snow depth and snow water equivalent (SWE) mean absolute errors (MAE), and slightly reduces snow cover, thereby better matching satellite-based snow cover observations. Compared to an open loop (no DA) experiment (OL), and an experiment with an assumed static observation error ($DA_{const}$), $DA_{var}$ reduces SWE MAE by 25% and 13%, respectively, compared with over 8000 manual SWE measurements. This work demonstrates the benefits of machine learning based snow depth retrievals and the impact of incorporating dynamic observation errors in EnKF-based snow DA.

## 1 Introduction

Snow is a valuable natural resource, integral for societal needs and in the climate system. The runoff from seasonal snow serves as a water source for billions of people (Barnett et al., 2005; Mankin et al., 2015), supports clean hydroelectric energy generation (Wasti et al., 2022), and sustains irrigated agriculture (Qin et al., 2020). Snow is also necessary for the multi-billion

dollar winter tourism industry (Outdoor Industry Association, 2017; Parthum and Christensen, 2022; Steiger et al., 2019). The total economic value of snow is estimated to be in the trillions of dollars (Sturm et al., 2017). Furthermore, snow has a high albedo and therefore plays an important role within the climate system by exerting a large-scale cooling effect. Variability in snow cover therefore impacts the Earth's surface energy balance and has been shown to potentially affect Northern Hemisphere atmospheric circulation (Henderson et al., 2018). Significant changes including a decline in snow-covered area, particularly at low elevations (Bormann et al., 2018; Estilow et al., 2015), shifts in the timing of snow melt (Musselman et al., 2021; Vorkauf et al., 2021), and an increasing transition from snowfall to rainfall at lower elevations (Safeeq et al., 2016) have been observed in recent decades, with these changes projected to intensify throughout the 21$^{st}$ century (IPCC, 2021).

Despite the importance of snow within Earth's climate and as a natural resource, accurately quantifying snow mass (or snow water equivalent, SWE) in mountainous, complex terrain remains a challenge. Because SWE is difficult and costly to directly quantify (Dozier et al., 2016), measurements and retrieval algorithms more commonly focus on snow depth, which is related to SWE via snow density. In-situ observation stations provide point-based snow depth measurements with good temporal frequency, but fail to capture spatial snow variability, which can be great even in a small area (López-Moreno et al., 2015; Miller et al., 2022). Airborne surveys provide accurate snow depth maps at a fine spatial resolution (Deems et al., 2013), but their high costs and logistical constraints limit the frequency and spatial coverage of these measurements. Snow depth has also been retrieved using satellite observations, which have the benefit of providing frequent, global coverage (Lievens et al., 2019). One approach estimates snow depth by comparing digital elevation models (DEMs) from snow-on and snow-off conditions. These DEMs can be generated from satellite laser altimetry such as ICESat-2 (Enderlin et al., 2022; Deschamps-Berger et al., 2023; Besso et al., 2024) or from very-high-resolution stereoscopic satellite imagery via photogrammetric methods (Marti et al., 2016; Shaw et al., 2020; Deschamps-Berger et al., 2020). Globally, passive microwave and synthetic aperture radar (SAR) observations are more commonly used to estimate snow depth. (Kelly et al., 2019; Luojus et al., 2021; Lievens et al., 2022). However, passive microwave imagery has a coarse spatial resolution ($\sim$25 km) and saturates above 1 m snow depth (Tedesco and Narvekar, 2010; Vander Jagt et al., 2013), while SAR observations are challenged by wet snow, shallow snow, and forest cover (Broxton et al., 2024; Hoppinen et al., 2024; Lievens et al., 2022). Although recent work has utilized machine learning (ML) techniques to enhance SAR-based snow depth retrievals (Daudt et al., 2023; Broxton et al., 2024; Dunmire et al., 2024), there is still some way to go for accurate global SWE estimation.

Ultimately, complex feedbacks between changes in snow and other components of the global climate system are currently best studied using physics-based models (Girotto et al., 2020). Since in-situ SWE observations are far sparser than snow depth measurements (Dunmire et al., 2024), snow mass estimates also rely primarily on modeling approaches. However, these models are limited by uncertainties in mountain precipitation and low-quality forcing data (Günther et al., 2019; Raleigh et al., 2016; Terzago et al., 2020). In light of these observational and modeling challenges, data assimilation (DA) offers a way to overcome shortcomings of both the model and observations by integrating in-situ and remote satellite observations with physics-based models to improve modeled snow variables (Helmert et al., 2018; Smyth et al., 2020, 2022).

One method for assimilating observations into a physical model is via direct insertion, whereby the model's state variables are directly replaced with observations without any statistical blending or error weighting (Rodell and Houser, 2004; Toure

et al., 2018). Increasing in sophistication, optimal interpolation methods, which consider model and observational uncertainty to blend the model and observations using statistically optimal weights (Liston and Hiemstra, 2008), are commonly used at operational centers (Helmert et al., 2018). Also common among operational centers (Helmert et al., 2018), and one of the most used DA techniques within the land surface modeling community, is the Ensemble Kalman Filter (EnKF; Reichle et al. (2002)). With an EnKF, the background-error covariance is not explicitly computed, but instead estimated using an ensemble of model trajectories. While this ensemble approach is advantageous for high-dimensional, nonlinear systems where an exact computation of the background-error covariance is impractical, the assumption of unbiased, normally distributed model-state errors is often violated for cumulative state variables like snow depth. Despite its reliance on Gaussian assumptions, the EnKF has been extensively used in previous snow data assimilation work (Slater and Clark, 2006; Durand and Margulis, 2006; De Lannoy et al., 2012; Huang et al., 2017; Pflug et al., 2024). An alternative solution that is commonly used in snow DA, particle batch filters and smoothers are capable of handling non-Gaussian noise and complex posterior distributions. In particular, particle batch smoothers have been commonly applied to create snow reconstructions (Margulis et al., 2015; Baldo and Margulis, 2018) or to downscale model variables such as precipitation (Girotto et al., 2024; Bachand et al., 2025).

Recent studies have used both particle batch smoothers and the EnKF to assimilate SAR-based snow depth retrievals from Sentinel-1 (S1), thereby improving modeled snow depth, SWE and streamflow compared to in-situ measurements (De Lannoy et al., 2024; Brangers et al., 2024; Girotto et al., 2024; Mirza et al., 2025). However, these previous snow DA studies make the simplifying assumption that the observation uncertainty is constant in space and time, meaning that a 10 cm snowpack is assumed to have the same absolute uncertainty as a 400 cm snowpack, contributing to a suboptimal use of the observational information.

Here, we present several advances in snow DA. First, we assimilate snow depth retrievals from an ML product that uses S1 observations, land cover, and topographic information to estimate snow depth in the European Alps (Dunmire et al., 2024). These ML-based snow depth retrievals have a higher accuracy and lower bias compared to previous S1-based retrievals from a conceptual model (Lievens et al., 2022), when validated against in-situ observations and airborne snow depth maps from the European Alps. For instance, compared to 798 Alps-wide in-situ measurement sites, the ML model has an average site mean absolute error (MAE) of 0.18 m and an average site bias of -8 mm, compared to an MAE of 0.22 m and a bias of -99 mm for the conceptual model, respectively. We assimilate these ML-based snow depth retrievals within a land surface model over the entire European Alps, a domain much larger than most previous snow DA efforts which focus primarily on smaller, regional scales. Finally, we incorporate a dynamic observation error, whereby the uncertainty of the assimilated snow depth observation varies in space and time, reflecting the more realistic dynamics of uncertainty in snowpack observations. The primary goal of this work is to assess the utility of incorporating dynamic observation errors versus commonly used static observation errors in EnKF-based snow DA.

## 2 Materials and methodology

In this work, we utilized the NASA Land Information System (LIS; Kumar et al. (2006); Peters-Lidard et al. (2007)) version 7.5.0 to assimilate snow depth retrievals in the Noah-MP land surface model (Niu et al., 2011; Yang et al., 2011) version 4.0.1. The snow depth retrievals, land surface model, DA experiments, and evaluation data and methods are further described below.

### 2.1 Model setup and data

#### 2.1.1 Noah-MP land surface model

To simulate snow processes over the European Alps (3.9945°E–17.0175°E, 42.9945°N–48.6195°N), we ran Noah-MP on a regular latitude-longitude grid with a spatial resolution of 0.009°. In Noah-MP, snow is simulated in up to 3 layers, depending on the total snow depth. Snow processes and properties such as melt metamorphism, canopy interception, and snow cover fraction are represented by detailed physically-based parameterizations (Niu et al., 2011). For snow albedo, we used the Canadian Land Surface Scheme (CLASS; Verseghy (1991)). For other parameterization options, we followed Brangers et al. (2024).

Before beginning our DA experiments, we performed a 15-year model spin-up (2000-2015). The experiments were conducted over the period spanning October 1, 2015 – April 30, 2023 (8 snow seasons). Noah-MP was run with a 15 minute model time step and daily averages of state variables were written to output.

#### 2.1.2 Atmospheric forcing for Noah-MP

The model was forced with atmospheric forcing from the ECMWF Reanalysis, version 5 (ERA5; Hersbach et al. (2020)). The ERA5 data were downscaled from their native resolution (31 km) to the domain grid through bilinear spatial interpolation and by applying a topographic lapse-rate correction to correct the air-temperature forcing. ERA5 has previously been used as atmospheric forcing in other snow DA studies (Pflug et al., 2024; De Lannoy et al., 2024; Mirza et al., 2025), and Brangers et al. (2024) additionally demonstrated that ERA5 forcing leads to superior modeled snow depth, compared with simulations forced with The Modern-Era Retrospective Analysis for Research and Applications, version 2 (MERRA-2; Gelaro et al. (2017)), and MERRA-2 gauge-corrected precipitation (M2CORR; Reichle et al. (2017)). From Figure 10 of Brangers et al. (2024), the ERA5, MERRA-2, and M2CORR atmospheric forcing led to average modeled snow depth MAEs of 0.367 m, 0.404 m, and 0.434 m, and average snow depth biases of -0.07 m, +0.138 m, and -0.363 m, respectively, compared to in-situ measurement stations in the Western European Alps.

### 2.2 Machine learning snow depth retrieval

Previous work has assimilated snow depths retrieved from the S1 satellite constellation over the European Alps ($SD_{S1}$; Brangers et al. (2024); De Lannoy et al. (2024)). Here, we assimilated snow depth estimates from Dunmire et al. (2024) ($SD_{ML}$), which uses machine learning to enhance S1-based snow depth retrievals. Dunmire et al. (2024) use an eXtreme Gradient Boosting (XGBoost) model that incorporates 12 input features (elevation, slope, aspect angle, topographical posi-

**Table 1.** Perturbation parameters applied for the OL and DA runs. * We perturb the total snow depth and propagate these perturbations into the different snow layers.

| Variable | Perturbation type | Standard deviation | Cross-correlation | | | |
|---|---|---|---|---|---|---|
| Forcing variables | | | SW | LW | P | T |
| SW: Incident shortwave ($\mathrm{W\,m^{-2}}$) | multiplicative | 0.6 | 1 | -0.5 | -0.5 | 0.3 |
| LW: Incident longwave ($\mathrm{W\,m^{-2}}$) | additive | 50.0 | 0.5 | 1 | 0.5 | 0.6 |
| P: Precipitation ($\mathrm{kg\,m^{-2}\,s^{-2}}$) | multiplicative | 0.5 | -0.5 | 0.5 | 1 | -0.1 |
| T: 2 m air temperature (K) | additive | 1.0 | 0.3 | 0.6 | -0.1 | 1 |
| Forecast variable* | | | | | | |
| Snow depth (m) | multiplicative | 0.0005 | | | | |

tion index, snow class, forest cover fraction, day of snow season, snow cover fraction, cumulative snow cover fraction, local incidence angle of the S1 observation, S1 VV backscatter, and S1 cross-polarization ratio) to estimate snow depth across the European Alps at 100 m resolution. When compared to in-situ snow depth stations and airborne photogrammetry snow depth maps, $SD_{ML}$ is shown to reduce MAE and improve bias compared to $SD_{S1}$ (MAE reduction from 0.22 m for $SD_{S1}$ to 0.18 m for $SD_{ML}$, bias improvement from -99 mm for $SD_{S1}$ to -8 mm for $SD_{ML}$) (Dunmire et al., 2024; Lievens et al., 2022).


We spatially averaged the $SD_{ML}$ retrievals to the 0.009° model resolution and masked pixels with a glacier fraction above 50%, according to version 7 of the Randolph Glacier Inventory (Pfeffer et al., 2014; RGI 7.0 Consortium, 2023). We also temporally averaged the $SD_{ML}$ retrievals every 7 days and assimilated these estimates weekly, in the center of the 7-day averaging window. This step was taken to avoid assimilating outlier snow depths (the $SD_{ML}$ retrievals can be noisy in time) and to avoid negative consequences (e.g. spurious temporal trends) associated with a changing assimilation frequency (Dee, 2005).


## 2.3 Data assimilation approach and experiments

We conducted 3 different experiments: (1) an open loop, model-only experiment (OL) which serves as a benchmark to evaluate the added value of assimilating $SD_{ML}$ retrievals, (2) a DA experiment with an assumed constant observation error ($\mathrm{DA_{const}}$), and (3) a DA experiment with a dynamic observation error that varies spatially and temporally ($\mathrm{DA_{var}}$). For all experiments, we utilized 12 ensemble members, created by perturbing forcing variables (precipitation, 2 m air temperature, and incident longwave and shortwave radiation) and the total forecasted snow depth (with the total snow depth perturbations distributed over the snow layers). Although a larger ensemble size is more optimal, our choice of 12 ensembles is reasonable as the control vector used in the assimilation consists of just total snow depth (Pflug et al., 2024). The perturbation parameters are summarized in Table 1 and follow Modanesi et al. (2022), Bechtold et al. (2023), and Pflug et al. (2024).



For the DA experiments, we used a one-dimensional EnKF to assimilate the $SD_{ML}$ retrievals into Noah-MP. The Kalman gain matrix determines the strength of the model corrections at each location ($x$) and timestep ($t$), and is given by Equation 1 below:

$$K(x,t) = \frac{\sigma_f^2(x,t)}{\sigma_f^2(x,t) + \sigma_{obs}^2} \tag{1}$$

where $\sigma_f$ is the standard deviation of the forecast error and represents the uncertainty in the forecast's total snow, and $\sigma_{obs}$ is the standard deviation of the observation error and represents the uncertainty in the observations. The EnKF extends a traditional Kalman Filter by estimating $\sigma_f$ using forecast ensembles, while $\sigma_{obs}$ is a user-defined parameter. Here, we tested two different approaches for $\sigma_{obs}$, one that is constant (DA$_{const}$) and one that varies in space and time (DA$_{var}$).

As per De Lannoy et al. (2024), the DA$_{const}$ experiment assumes a constant value of $\sigma_{obs} = 0.3$ m. The multiplicative factor

for the snow depth state perturbations (Table 1) was determined experimentally through trial and error, with the optimal value selected based on its performance compared to in-situ snow depth observations over a subset region (Brangers et al. (2024), personal communication, Isis Brangers).

The DA$_{var}$ experiment expands upon DA$_{const}$ by varying $\sigma_{obs}$ throughout space ($x$) and time ($t$) following Equation 2 below:

$$\sigma_{obs}(x,t) = \begin{cases} 0.05, & SD_{ML}(x,t) \leq 0.167, \\ m * SD_{ML}(x,t), & 0.167 < SD_{ML}(x,t) < 3.5, \\ 1.05, & SD_{ML}(x,t) \geq 3.5. \end{cases} \tag{2}$$

where $SD_{ML}(x,t)$ is the assimilated observation at location $x$ and time $t$ and $m$ is a user-defined multiplier. We calibrated $m$ experimentally by selecting the optimal value when comparing modeled snow depth with in-situ observations in a subset region (6-8 °E, 45-46 °N). Here, we used $m = 0.3$. Equation 2 assumes that $\sigma_{obs}$ varies linearly as a function of assimilated snow depth. Supplemental Figure 1 demonstrates that this assumption is valid at independent in-situ measurement sites. For

$SD_{ML}$ below 0.25 m, the average error of the $SD_{ML}$ product compared to in-situ measurements is 0.05 (Supplemental Figure 1), and as such we chose this as a minimum threshold value for $\sigma_{obs}$ (Equation 2). Setting this minimum threshold also avoids issues when $SD_{ML}(i,t) = 0$ m. We can see from Supplemental Figure 1 that there are no assimilated snow depths above 3 m at these in-situ measurement sites, making it difficult to characterize the observation error for deeper assimilated snow depths. As such, we also defined an upper threshold for $\sigma_{obs}$ of 1.05 m, corresponding to an assimilated snow depth of 3.5 m (Equation

2). This value was also chosen as an upper threshold because we observed that $\sigma_f$, which represents the uncertainty in the model-only (OL) simulated snow depth, given by the standard deviation of the model ensembles, levels off above 3.5 m snow depth (Supplemental Figure 2). We chose to reflect this feature of the forecast error in our characterization of the observation error. Figure 1 compares $\sigma_{obs}$ from DA$_{const}$ and DA$_{var}$ as a function of the assimilated snow depth observation ($SD_{ML}$).

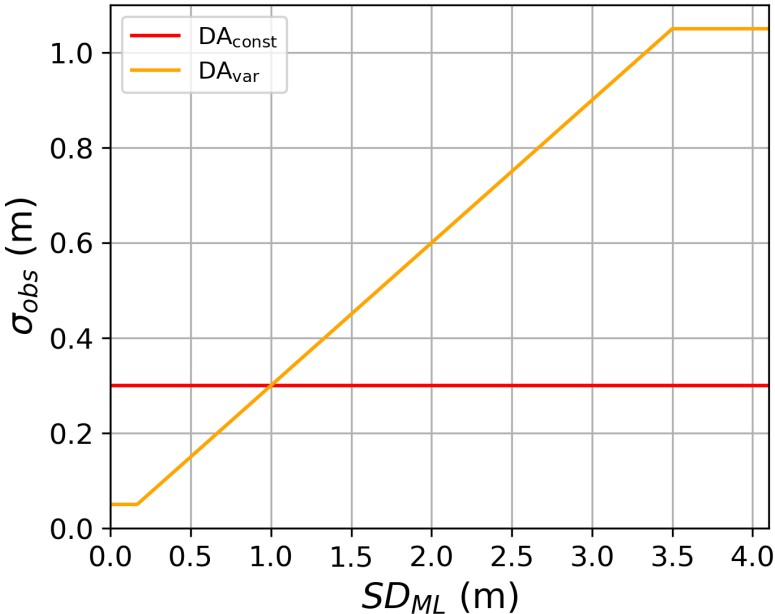

**Figure 1.** Assumed observation error standard deviation ($\sigma_{obs}$) as a function of the assimilated snow depth ($SD_{ML}$) for the two DA experiments.

For both DA experiments, the snow updates were applied following the methodology of Brangers et al. (2024), whereby the increments applied to the total forecasted snow depth are divided over the different snow layers, proportionate to each layer's forecasted share of the total snowpack, and SWE is updated accordingly assuming snow density remains unchanged during each update. The compaction and redistribution of snow layers is done during the model propagation. This approach circumvents the need to compute dynamic error covariances between total snow depth and a varying number of snow state variables in varying numbers of layers. We assimilated $SD_{ML}$ estimates weekly each year from September 1 through March 31, excluding assimilation further into the ablation period when wet snow complicates the S1 signal. Due to limitations of using S1 observations to estimate snow depth in forested terrain, and the unsuitability of the ML SD retrieval over glaciated terrain, we do not assimilate over forested areas or glaciers. Following De Lannoy et al. (2024), we also do not assimilate when the soil or vegetation temperature is above 5°C.

## 2.4 Evaluation

For each of our three experiments (OL, DA$_{const}$, DA$_{var}$), we utilized a variety of in-situ and satellite-based products to evaluate 1) snow depth, 2) SWE, and 3) snow cover fraction (SCF) and snow disappearance date (SDD). We also compared our results with those from De Lannoy et al. (2024), in which the $SD_{S1}$ retrieval was assimilated with a static observation uncertainty.

### 2.4.1 Snow depth evaluation

Snow depth estimates from each experiment (OL, $DA_{const}$, $DA_{var}$) were compared with in-situ snow depth observations from across the European Alps. For comparing the performance of our DA experiments against the OL experiment, we utilized independent in-situ observations that were not included in the training for the ML model from Dunmire et al. (2024), and sites located in places where the $SD_{ML}$ retrievals were assimilated (i.e. not in dense forest, over glaciers). In total, we utilized snow depth data from 588 measurement sites, which report for varying parts of the 8 year study period. We obtained these point-scale snow depth measurements from the WSL – Institute for Snow and Avalanche Research SLF (Switzerland, 220 sites), Météo-France (France, 57 sites), GeoSphere Austria (Austria, 108 sites), the International Center for Environmental Monitoring CIMA Research Foundation (Italy, 10 sites), Provincia autonoma di Trento (Italy, 48 sites), Provincia autonoma di Bolzano - Alto Adige (Italy, 19 sites), Valle d'Aosta (Italy, 27 sites), the Agenzia Regionale per la Protezione Ambientale - Piemonte (Italy, 28 sites), the European Centre for Medium-Range Weather Forecasts' SYNOP snow depth measurement network (Global, 35 sites; de Rosnay et al. (2015)), and Global Historical Climatology Network (Global, 36 sites). For each experiment, we computed the mean absolute error (MAE), bias, and Pearson correlation coefficient (R) of the modeled snow depth compared with the in-situ observations obtained at these sites. To investigate how well the model captures spatial and temporal anomalies in snow depth patterns, we also computed spatial and temporal anomaly correlation coefficients (ACC). The spatial ACC was computed for each day throughout the snow season with more than 10 in-situ snow depth measurements available. Spatial anomalies were computed for each site by subtracting the spatial mean snow depth recorded across all measurement sites on that day. The temporal ACC was computed for each measurement site with 5 or more years of in-situ observations. Temporal anomalies were calculated at each site by subtracting the site's multi-year climatology (2015-2023) with a 10-day moving mean smoothing function applied. In order to utilize more sites with a longer time series of observations, we also included sites that were used in the ML training. Thus, for this metric, we only compared the two DA experiments, which both assimilated the same $SD_{ML}$ retrievals.

### 2.4.2 SWE evaluation

Next, we evaluated modeled SWE, with in-situ measurements of SWE located (1) in places where DA was applied, and (2) not on a glacier, according to the Noah-MP glacier land cover class and the Randolph Glacier Inventory (Pfeffer et al., 2014; RGI 7.0 Consortium, 2023). We consolidated 8211 manual SWE measurements from the Bundesministerium für Land- und Forstwirtschaft, Regionen und Wasserwirtschaft (Austria, 676 measurements), The Climate Data Center of the German Weather Service (Germany, 2311 measurements), the WSL – Institute for Snow and Avalanche Research SLF (Switzerland, 1546 measurements), Provincia autonoma di Trento (Italy, 944 measurements), and Valle d'Aosta (Italy, 2793 measurements). As with snow depth, we compared MAE, bias, and R for the different experiments.

### 2.4.3 SCF and SDD evaluation

We further evaluated the impact of the DA on the timing of snow disappearance and modeled SCF. We first compared the SDD of the model experiments at the in-situ snow measurement sites. We defined the SDD as the first day of five consecutive days with less than 0.1 mm snow depth, following the date of peak snow. For in-situ SDD, the day of peak snow was computed using the in-situ snow depth and for model SDD, the day of peak snow was computed using snow depth output from the appropriate model experiment. We also, in the same manner, computed SDD using the Interactive Multisensor Snow and Ice Mapping System (IMS) product. IMS is a 1 km horizontal resolution binary snow cover dataset that is derived from a variety of satellite and in-situ data.

We also compared SCF and total snow covered area from our three model experiments with both the IMS product (U.S. National Ice Center, 2008) and the Copernicus Fractional Snow Cover product (European Union's Copernicus Land Monitoring Service information). The Copernicus product is available at a 20 m spatial resolution and is computed from Sentinel-2 Level-1C imagery. The product is not gap-filled, thus data gaps exist when clouds are present. We regridded both snow cover products to our model domain grid using nearest neighbor interpolation for IMS, and averaging for the Copernicus product. For comparison with the IMS product, we converted modeled SCF to a binary value: SCF $< 50\% = 0$, SCF $\geq 50\% = 1$. For comparison with the Copernicus product, we ignored areas with data gaps.

### 2.4.4 Comparison to $SD_{S1}$ DA

To compare with previous work that assimilates snow depth retrievals from the S1 change detection algorithm ($SD_{S1}$; Lievens et al. (2022)), we compared output from our two DA experiments with DA output from De Lannoy et al. (2024) (experiment $DA_{S1}$). This $DA_{S1}$ experiment utilized the same DA setup as in $DA_{const}$, with a static observation uncertainty ($\sigma_{obs} = 0.3\,m$), but assimilates $SD_{S1}$ retrievals instead of $SD_{ML}$. Here, we utilized 4548 manual SWE measurements collected within the Po River basin (the study domain of De Lannoy et al. (2024)) to compare SWE MAE between the $DA_{const}$, $DA_{var}$, and $DA_{S1}$ experiments.

## 3 Results

### 3.1 Snow depth

The practical impact of the $DA_{var}$ and $DA_{const}$ experiments on snow depth estimates is illustrated in Figure 2. When the assimilated snow depth retrieval is 1 m, the observation uncertainty is equivalent for both experiments (Fig. 1). The variable observation uncertainty approach in $DA_{var}$ dynamically adapts to assimilated snow depth, resulting in stronger corrections for shallow depths while $DA_{const}$ provides stronger corrections at higher depths (Fig. 2). For assimilated snow depths below 1 m, the observation uncertainty is smaller in $DA_{var}$ than in $DA_{const}$, resulting in a lower observation error covariance ($\sigma_{obs}$) in the EnKF (Equation 1) and stronger corrections of the posterior state toward the observations in $DA_{var}$ (Fig. 2a). In contrast, the

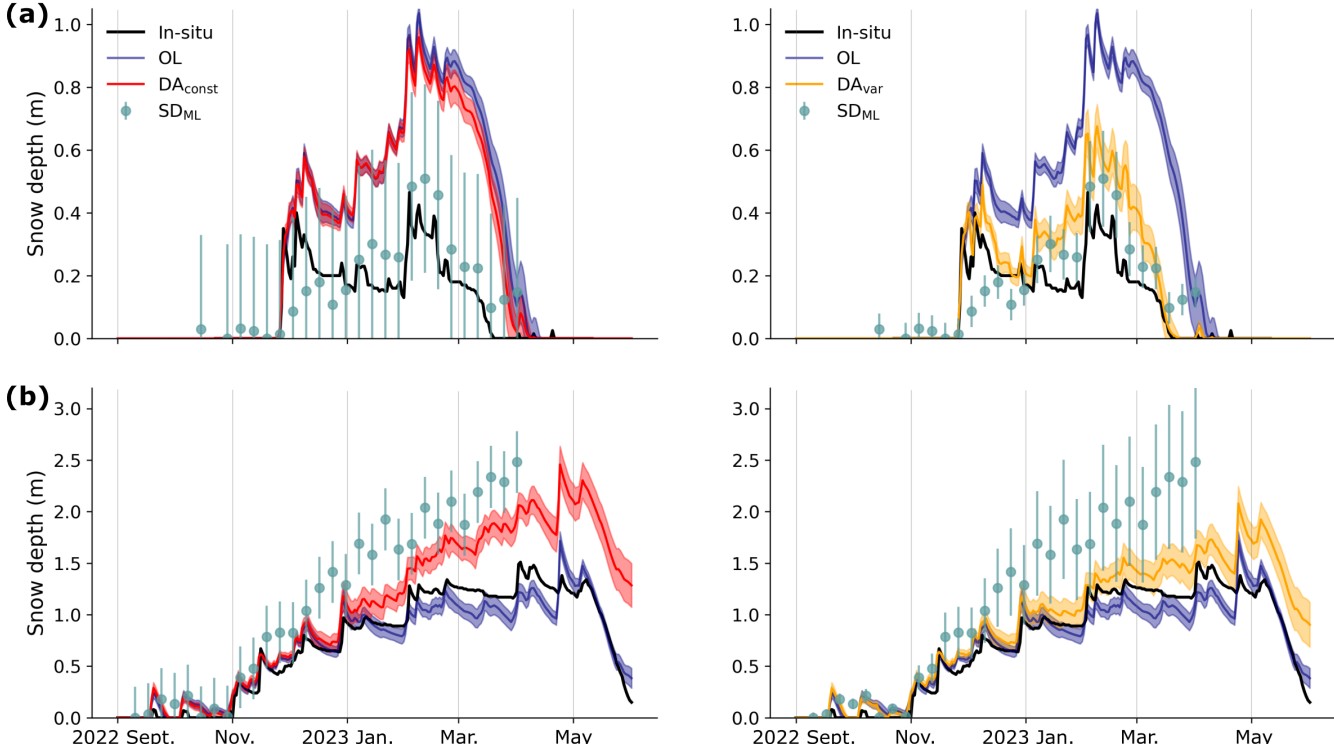

**Figure 2.** Snow depth estimates and independent in-situ measurements at two example sites. (a) Snow depth from $DA_{const}$ (red, left) and $DA_{var}$ (orange, right) compared with the OL (navy) from a measurement station in Austria ($\sim$13.6228 $^{\circ}E$, 47.0944 $^{\circ}N$, 1050 m elevation). The shading represents $\pm$1 standard deviation in the model ensembles. The sage green dots represent the assimilated $SD_{ML}$ retrievals, with error bars for the assumed observation error standard deviation ($\sigma_{obs}$, Equation 2). (b) Same as (a), but for a different measurement station in Switzerland ($\sim$7.7836 $^{\circ}E$, 45.9872 $^{\circ}N$, 2948 m elevation). These two sites were chosen due to a lack of gaps in the in-situ measurements and their general representativeness of locations where the DA removes and adds snow.

constant $\sigma_{obs}$ of 0.3 m in $DA_{const}$ is relatively large for shallow snow depths and results in minimal corrections of the posterior state.

A measurement site with assimilated snow depths greater than 1 m is demonstrated in Figure 2b. In this case, the observation uncertainty is smaller for $DA_{const}$ than for $DA_{var}$, resulting in stronger posterior state adjustments in $DA_{const}$. At this measurement site we see that the OL experiment is closer to the in-situ snow depth than the assimilated observations, leading to a deterioration in model performance when the DA is applied (both with $DA_{var}$ and $DA_{const}$). For $DA_{var}$, this phenomenon occurs at $\sim$16% of all measurement site (Fig. 3a), with 1% experiencing a deterioration in SD MAE greater than 125 mm.

Across the 588 in-situ snow depth measurement sites used for evaluation, the corrections applied in $DA_{var}$ result in snow depth estimates that align more closely with in-situ observations (Fig. 3). The OL experiment yields a site-average MAE of 0.244 m, a RMSE of of 0.300 m, a bias of 0.113 m and a Pearson correlation coefficient of 0.75. Both the $DA_{const}$ and $DA_{var}$

experiments improve these metrics, with site-average MAE values of 0.237 m and 0.215 m (median values of 0.207 m and 0.185 m), RMSE values of 0.292 m and 0.268 m, and biases of 0.106 m and 0.055 m, respectively. These improvements are illustrated in Figure 3, which compares MAE from the $DA_{var}$ experiment with the OL experiment (Fig. 3a) and $DA_{const}$ (Fig. 3b). Relative to the OL, MAE is reduced in $DA_{var}$ by more than 25 mm at 245 sites (42%), while 92 sites (16%) have an MAE increase exceeding 25 mm. Comparing $DA_{var}$ to $DA_{const}$, we find that MAE is reduced in $DA_{var}$ by more than 15 mm at 297 sites (51%), while 71 sites (12%) experience a deterioration greater than 15 mm. While improvement in MAE from the OL experiment is not significant for $DA_{const}$ (Mann-Whitney U test p-value = 0.59, median-test p-value = 0.68), the MAE improvement is small, but significant for $DA_{var}$ (Mann-Whitney U test p-value = 0.001, median-test p-value = 0.03). The site-average Pearson correlation coefficient slightly deteriorated for $DA_{const}$ and improved for $DA_{var}$ to 0.75 and 0.76, respectively.

While the OL experiment already does a good job at representing spatial snow depth patterns (spatial ACC = 0.71), Figure 3c highlights that, for most of the snow season, the $DA_{var}$ experiment offers slight improvements in the representation of these spatial patterns. Averaged across the entire year, the spatial ACC increases from 0.71 for the OL experiment to 0.72 for $DA_{const}$ and to 0.73 for $DA_{var}$. The greatest improvement in spatial ACC for $DA_{var}$ occurs during the early snow season (November), with values exceeding those of the OL and $DA_{const}$ experiments by 0.058 and 0.047, respectively. From December through April, the spatial ACC for $DA_{var}$ remains approximately 0.021 greater than that of the OL experiment. By mid-April, all three experiments exhibit similar performance in capturing spatial snow depth patterns. Additionally, both $DA_{const}$ and $DA_{var}$ well-capture temporal snow depth patterns, with average temporal ACC values of 0.68 and 0.72, respectively (median temporal ACC values of 0.73 and 0.76, respectively). The improvement in temporal ACC for $DA_{var}$ from both the OL and $DA_{const}$ is statistically significant (p < 0.01 for both a Mann-Whitney U test and median-test, Fig. 3d). Across the 948 sites evaluated, 491 sites (52%) have an improved temporal ACC in $DA_{var}$ (> +0.02 compared to $DA_{const}$), while 103 sites (11%) experience a deterioration in temporal ACC (< -0.02 compared to $DA_{const}$).

The OL experiment has an elevation-dependent snow depth bias, characterized by an overestimation of snow depth at lower elevations and early in the snow season, and an underestimation at higher elevations during peak snow accumulation (Fig. 4a). Both of these issues are mitigated in the $DA_{var}$ experiment, which brings seasonal biases closer to zero across all elevation bands (Fig. 4c). In contrast, the $DA_{const}$ experiment minimally corrects snow depth overestimation in the early season and at low elevations, due to the relatively higher assumed observation uncertainty for shallow snow (e.g., Fig. 2a). From September 1 through January 31, the $DA_{var}$ experiment reduces the average bias across all sites by 46%, while the $DA_{const}$ experiment achieves a 10% reduction over the same period. These improvements are particularly notable at mid-elevations (1000–2000 m), where $DA_{var}$ reduces model bias by 54% throughout the season, compared to a 13% reduction in model bias at these same sites in $DA_{const}$.

The MAE is also reduced by $DA_{var}$ across most elevation bands and throughout much of the season. The difference in MAE between the OL and $DA_{var}$ experiments (Fig. 4e) indicate that the largest MAE improvements occur from early winter through peak accumulation. However, an increase in MAE at high elevations during the melt season (March onwards) suggests

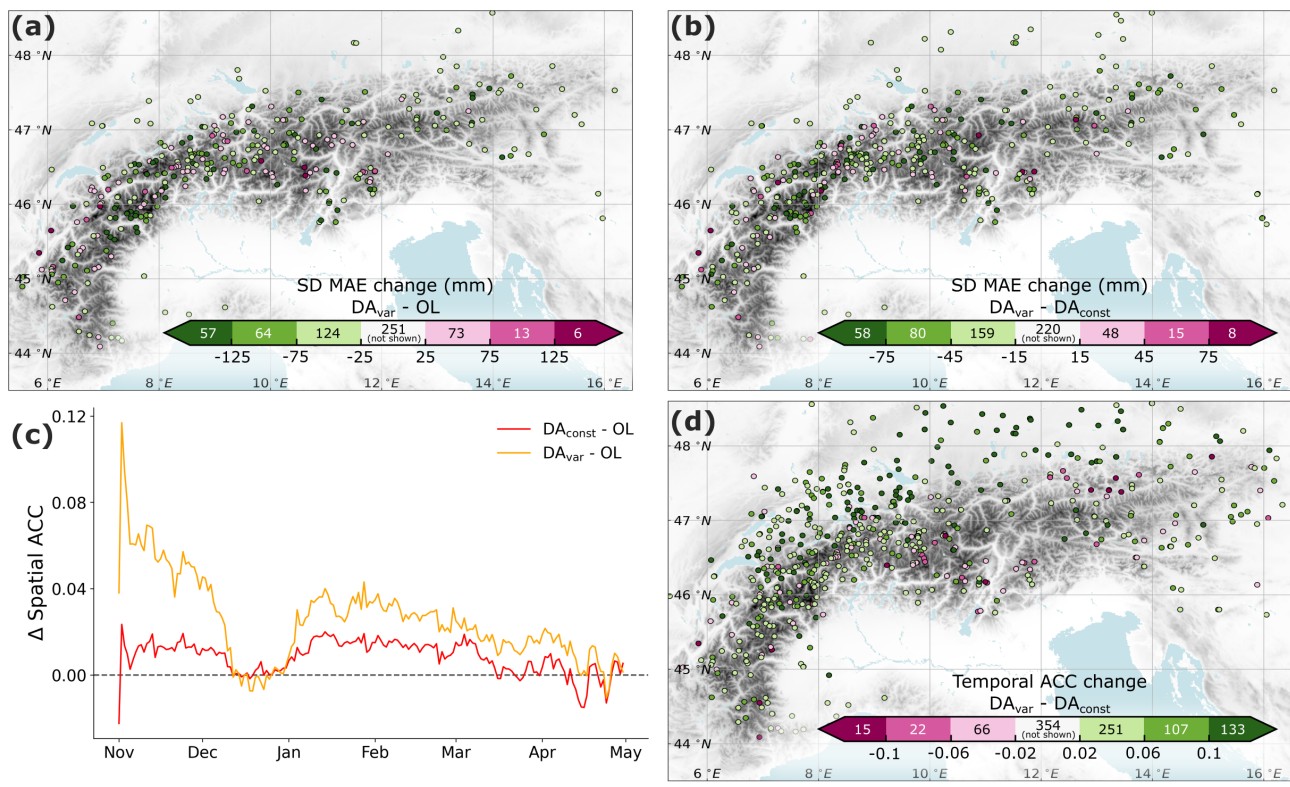

**Figure 3.** Experiment evaluation at in-situ snow depth measurement sites. (a) Change in MAE at each measurement site from the OL experiment to $DA_{var}$. Green colors indicate an improvement in MAE in the $DA_{var}$ experiment. On the color bar, the number of sites that fall within each color range is indicated and points within the white color are not plotted on the map. (b) Change in MAE at each measurement site from the $DA_{const}$ experiment to $DA_{var}$. (c) Change in the spatial anomaly correlation coefficient (ACC) for each DA experiment from the OL experiment. The spatial ACC is averaged over all snow seasons (2015/16 - 2022/23). (d) Change in the temporal ACC from the $DA_{const}$ experiment to $DA_{var}$. Green colors indicate an improvement in temporal ACC in the $DA_{var}$ experiment.

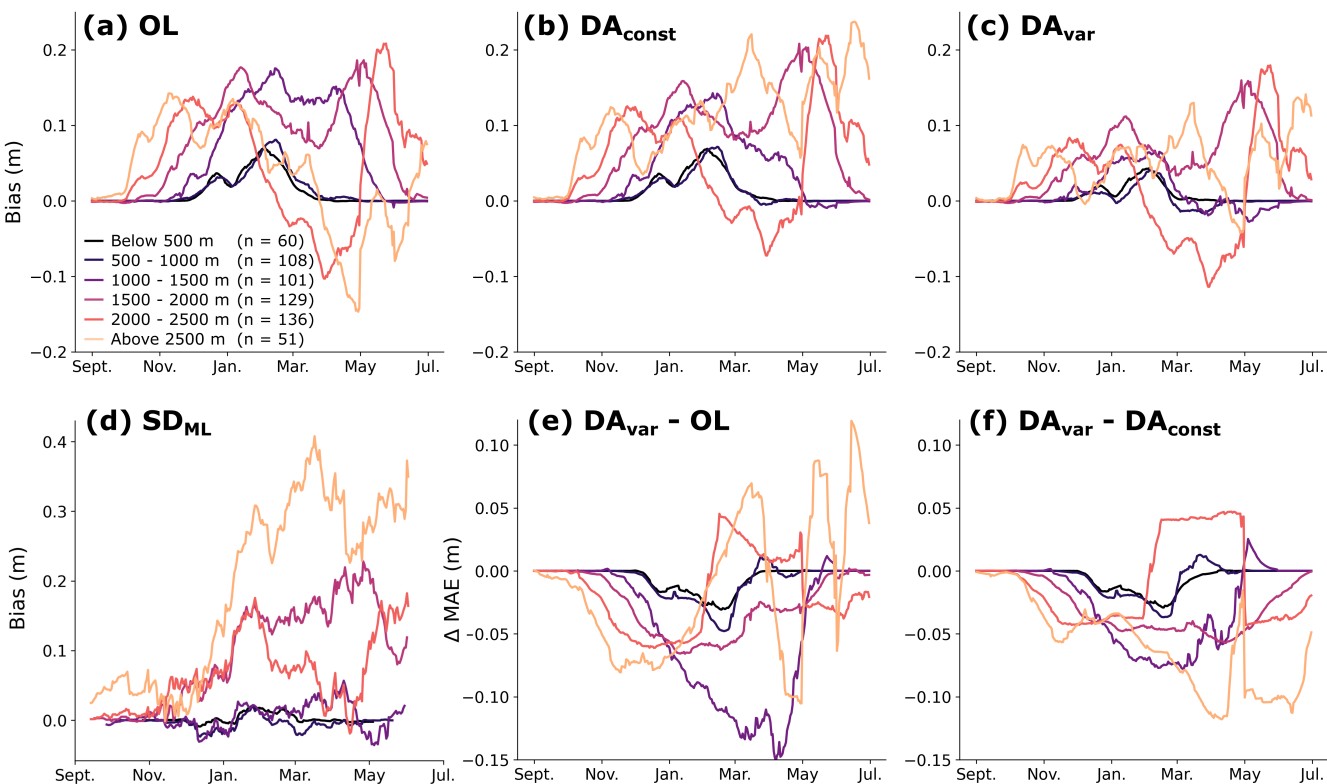

**Figure 4.** Seasonal evolution of bias and mean absolute error (MAE) stratified by elevation. Panels (a)-(d) show the seasonal snow depth bias for the (a) OL, (b) $DA_{const}$, and (c) $DA_{var}$ experiments, and for *(d) the assimilated observations ($SD_{ML}$)*. Bias is computed relative to in-situ snow depth measurements and is grouped by elevation bands (indicated by different colors). Panels (e)–(f) show the change in MAE between the OL and $DA_{var}$ experiments (e) and between the $DA_{const}$ and $DA_{var}$ experiments (f). Negative values in (e)–(f) indicate improved performance (decreased MAE). Statistics are computed for each day, averaged over the entire 8-year period (2015–2023). A 14-day smoothing is applied to each timeseries and the number of in-situ measurement sites (n) within each elevation band is provided in the legend.

a tendency for the DA experiments to retain snow for too long, which could be due to limitations in the modeled melt processes or biases introduced by the assimilated observations at higher elevations (e.g., Figure 4d).

## 3.2 SWE

Compared with 8,211 manual SWE measurements from 231 different measurement sites across the Alps, the $DA_{var}$ experiment also offers small, but significant improvements for SWE MAE compared to both the OL and $DA_{const}$ experiments (p<<0.001 for both a Mann-Whitney U test and median-test). Relative to the OL, $DA_{var}$ reduces SWE MAE by at least 15 mm at a majority of these sites (57%), while 23% of sites experience a deterioration in SWE MAE of more than 15 mm (Fig. 5a). Similar improvements are observed when comparing $DA_{var}$ to $DA_{const}$, with $DA_{var}$ outperforming $DA_{const}$ at 56% of measurement sites (Fig. 5b).

In the OL experiment, we observe a positive bias for low observed SWE and a negative bias for high observed SWE (Fig. 5c), similar to the bias patterns seen for snow depth. The $DA_{var}$ experiment reduces both biases, with the largest improvements occurring for low observed SWE values. For instance, for in-situ SWE below 200 mm, the bias is reduced by 52% in $DA_{var}$ compared to the OL (OL bias = +166 mm, $DA_{var}$ bias = +80 mm), meanwhile the bias in-situ SWE measurements above 600 mm is reduced by 7% in $DA_{var}$ (OL bias = -362 mm, $DA_{var}$ bias = -335 mm). As a result, the overall average SWE bias decreases from +81 mm in the OL to +18 mm in $DA_{var}$. In comparison, the bias reduction for the $DA_{const}$ (+76 mm bias) experiment is limited, because $DA_{const}$ marginally corrects the positive bias for low observed SWE, due to minimal model adjustments for shallow assimilated snow depths (e.g., Fig. 2a). Both $DA_{const}$ and $DA_{var}$ also improve the Pearson correlation coefficient (R = 0.60 for OL, R = 0.72 for $DA_{const}$, R = 0.71 for $DA_{var}$), indicating a stronger correlation with measured SWE.

Across all experiments, SWE typically peaks during the first week of March (March 1–7). Water Year 2017 recorded the lowest modeled SWE in our OL experiment, and correspondingly saw the largest SWE increases in $DA_{var}$ prior to early March, particularly in the Central Alps and Austrian Alps (Fig. 6a). However, $DA_{var}$ SWE improvements were mixed during this year. Of the 41 manual measurements taken between March 1 and March 7, 2017, only 24% demonstrated improved SWE MAE of more than 15 mm in $DA_{var}$. While the DA led to more accurately estimated SWE at some sites (e.g., Supplemental Fig. S3b,d), it resulted in an overestimation of SWE at others (e.g., Supplemental Fig. S3c,e,f). For example, three measurement sites in Italy (dark pink dots in Fig. 6a) experienced an average increase of 101 mm in added SWE in $DA_{var}$ relative to the OL. The average SWE MAE at these sites increased by 134 mm in $DA_{var}$, indicating that the assimilated $SD_{ML}$ observations overestimate snow at these locations. The degradation is even larger in $DA_{const}$, where the SWE MAE increases by 193 mm compared to the OL. This stronger deterioration arises from the lower assumed $\sigma_{obs}$ in $DA_{const}$ at these locations, which leads to stronger corrections toward the observations. A time series of modeled and observed SWE at one of these sites is shown in Supplemental Fig. S3e.

The largest SWE reductions from the OL to the $DA_{var}$ experiment occurred during Water Year 2018, particularly in the Bavarian Alps, Swiss Alps, and French Alps (Fig. 6b). In general, the reduced SWE in $DA_{var}$ aligns more closely with in-situ observations (e.g., Supplemental Fig. S4). The average SWE MAE for in-situ measurements taken between March 1-7, 2018 decreases from 164 mm in the OL, to 137 mm in $DA_{const}$ and 116 mm in $DA_{var}$. In $DA_{var}$, SWE MAE is improved by more than 15 mm in 59% of the 68 manual measurements taken between March 1 and March 7, 2018.

Water Year 2021 also experienced a large SWE reduction between the OL and $DA_{var}$ experiments, especially in the Swiss Alps and Eastern Dolomites. In the Dolomites region, where SWE reductions are often greater than 100 mm, a lack of in-situ observations makes it difficult to assess whether these reductions are realistic. However, limited measurement sites along the Italy-Austria border suggest that the SWE reductions may be too strong (e.g., Supplemental Fig. S5d). For instance, two in-situ measurements sites along the Italy-Austria border (indicated with yellow circles in Supplemental Fig. S5a) have an average SWE decrease of 142 mm in $DA_{var}$, and a corresponding degradation in SWE MAE of +113 mm. Meanwhile, southwest of these locations, eight measurement sites in Italy (black box in Supplemental Figure S5a) demonstrate contrasting improvements in $DA_{var}$ SWE MAE. At these 8 sites, SWE decreases by an average of 100 mm in $DA_{var}$, with a corresponding 74 mm

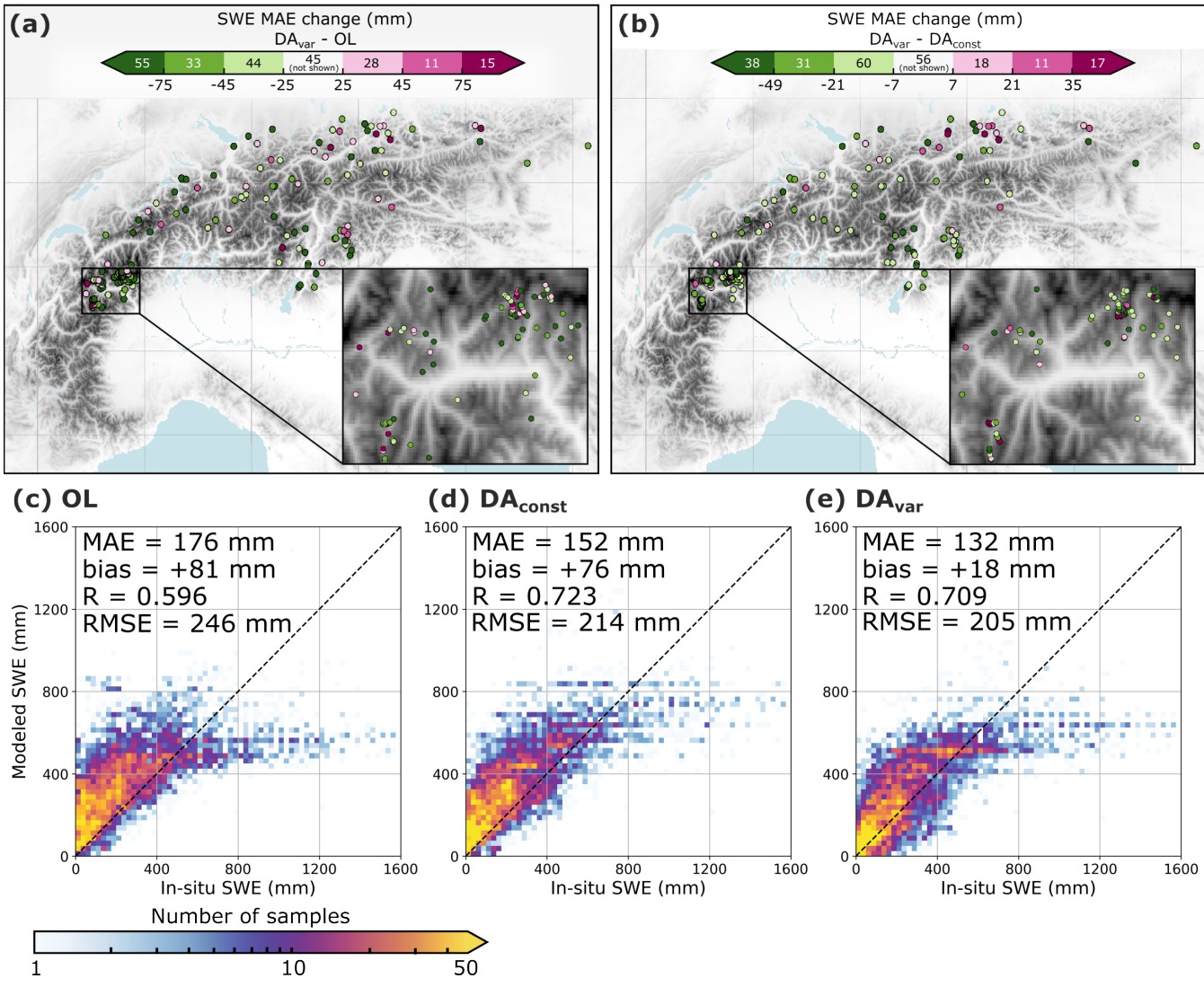

**Figure 5.** Evaluation of SWE from the OL and DA experiments against in-situ measurements. (a-b) Change in SWE MAE (a) $DA_{var}$ relative to the OL experiment, and (b) $DA_{var}$ relative to $DA_{const}$, where green indicates error reduction and magenta indicates a deterioration in performance. Measurements from within the same 1 km model grid cell are averaged for visualization purposes. On the color bar, the number of sites that fall within each color range is indicated and points within the white color are not shown on the map. (c-e) 2D histograms comparing modeled SWE to in-situ SWE observations for (c) OL, (d) $DA_{const}$, and (e) $DA_{var}$. All non-zero SWE measurements are included and the spatiotemporal MAE, bias, R, and RMSE are provided for each approach.

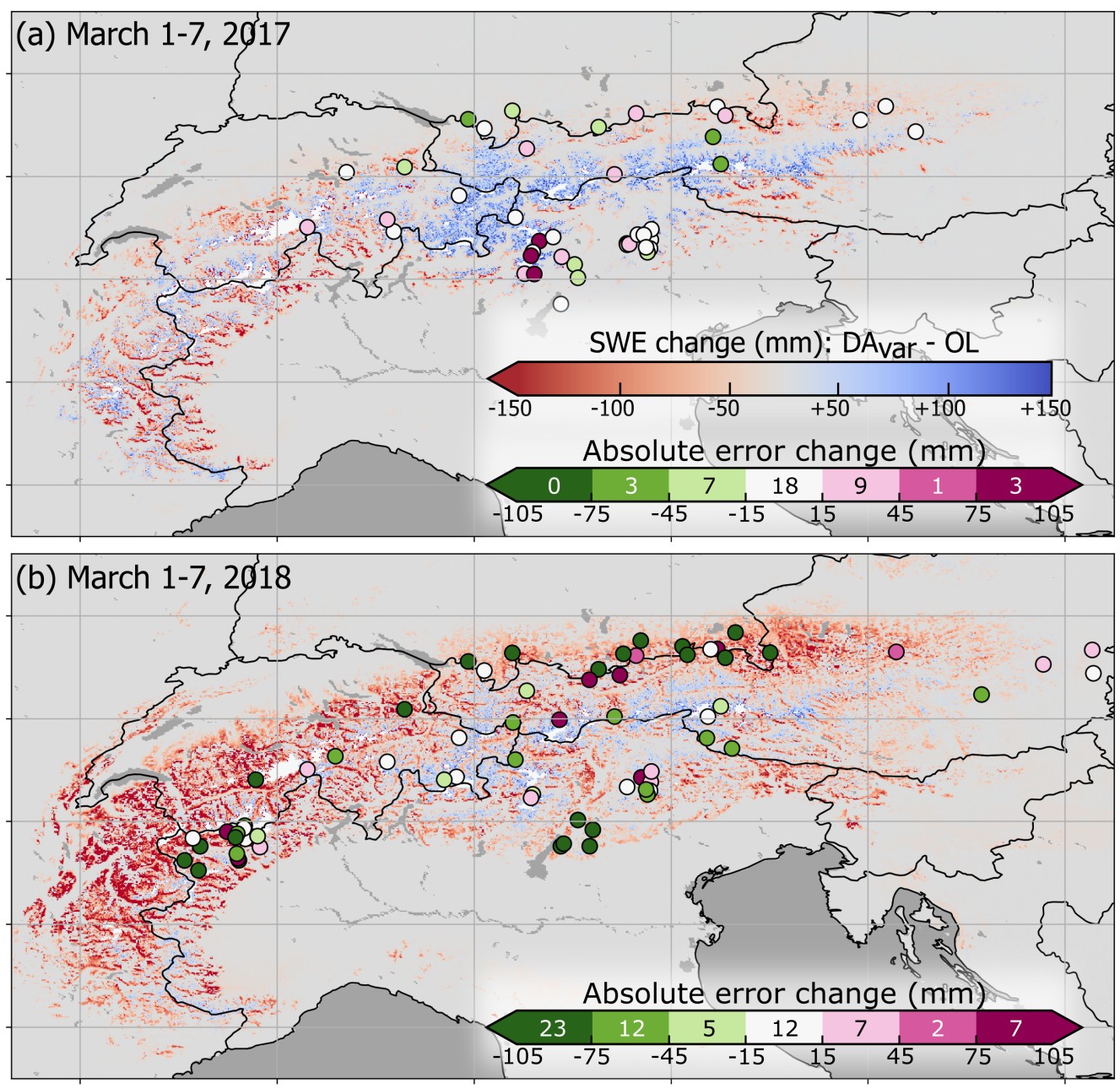

**Figure 6.** Change in SWE during the period March 1-7, between the $DA_{var}$ and OL experiments for (a) 2017 and (b) 2018. Manual SWE measurements taken during this period are plotted as dots, colored according to the change in absolute error between the $DA_{var}$ and OL experiments. On the error change color bar, the number of sites that fall within each color range is indicated.

reduction in SWE MAE. This result highlights some of the spatial inconsistencies of the DA improvements, which are likely due to spatial and temporal variation in the quality of the assimilated observations.

## 3.3 Snow cover and snow disappearance

The DA also affects snow cover estimates, contributing to a decrease in total snow-covered area leading up to peak snow accumulation in early March, and a slight increase in snow-covered area later in the season (April-May), compared with
335 the OL experiment (Fig. 7a). During peak snow accumulation in early March (March 1–7), the $DA_{var}$ experiment reduces total snow-covered area by 6,077 $km^2$ compared to the OL, averaged across the 2016-2023 period. Total snow-covered area during this same period in the $DA_{const}$ experiment is comparatively reduced by only 1,409 $km^2$. The relative difference in snow-covered area between $DA_{var}$ and the OL fluctuates more than for $DA_{const}$ (Fig. 7a), primarily due to the shallower early-season and low-elevation snowpacks in $DA_{var}$ which melt out more quickly.
The reduction in snow cover primarily occurs in low-elevation areas along the northern Alps (Fig. 7b), and aligns more closely with observed snow cover estimates from the IMS and Copernicus snow cover products, both of which indicate less snow-covered area than any of our model simulations. For example, on March 1, 2021, the OL and $DA_{var}$ experiments have, respectively, 79,345 $km^2$ and 58,091 $km^2$ more snow-covered area than the Copernicus fractional snow cover product, and 55,578 $km^2$ and 32,526 $km^2$ more than the IMS snow cover product (Supplemental Fig. S6). These discrepancies will be
discussed further in Section 4.
    At the majority of in-situ snow depth measurement sites, the estimated snow persists for too long compared to in-situ observations. Figure 8 presents cumulative distribution functions (CDFs), which show the cumulative number of sites with snow-free conditions after peak snow, stratified by elevation band. In all three model experiments, the snow disappearance date (SDD) occurs later than observed, indicating an overestimation of snow persistence across all elevation bands.
In $DA_{var}$, the SDD timing is improved at a majority of the observation sites located below 2000 m, with 51% of sites experiencing a SDD closer to in-situ observations, 22% experiencing a SDD farther from in-situ observations, and 27% remaining unchanged. The improvement is less pronounced for $DA_{const}$, in which 40% of sites show better agreement with observations, 24% show worse agreement, and 36% remain unchanged. The reduced SWE at lower elevations in $DA_{var}$ (see Section 3.2) likely results in more realistic timing for snow-free conditions at these sites. As we only assimilate observations through March,
thus limiting assimilation during times of ablation, changes in SDD are mainly a result in changes of peak SWE. In general, the IMS observations underestimate snow persistence (Fig. 8), leading to an earlier SDD compared to in-situ observations, which may result from the binary (as opposed to fractional) nature of the IMS observations.

## 3.4 DA increments and spread

In $DA_{const}$, model updates predominantly occur later in the accumulation season, with positive average increments above
360 2500 m and negative average increments below 1500 m (Fig. 9a). In contrast, $DA_{var}$ exhibits stronger negative increments earlier in the snow season, and at lower elevations (Fig. 9b), suggesting that assimilated observations influence the entire accumulation period rather than just times near peak SWE. Additionally, the magnitude of positive increments in $DA_{var}$ is

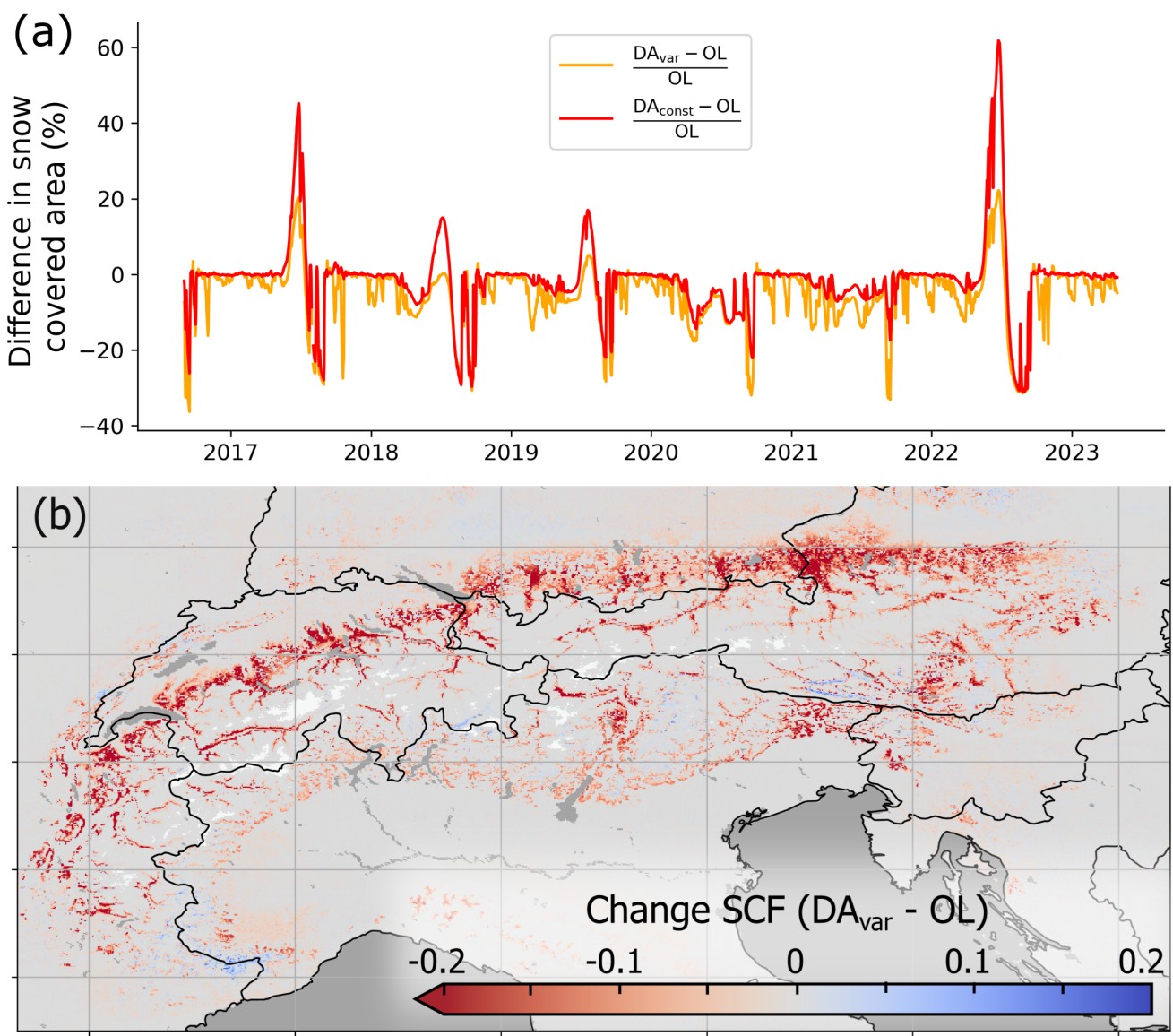

**Figure 7.** Difference in estimated snow covered area. (a) Timeseries of the percent difference in total snow covered area between $DA_{var}$ and OL (orange), and $DA_{const}$ and OL (red). (b) Average change in snow cover fraction (SCF) between the $DA_{var}$ and OL experiments during the period March 1-7 (all years). White indicates glaciers.

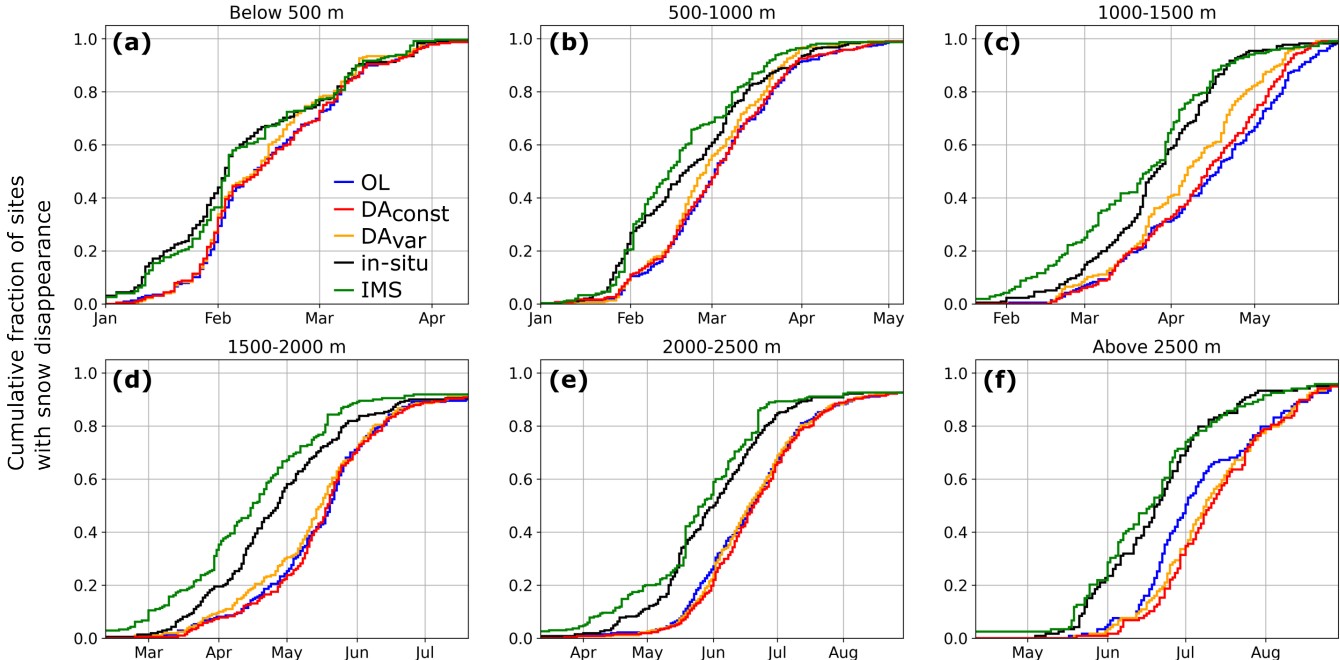

**Figure 8.** Cumulative fraction of measurement sites with snow disappearance (following peak snow depth) at in-situ measurement sites, stratified by elevation: (a) below 500 m, (b) 500-1000 m, (c) 1000-1500 m, (d) 1500-2000 m, (e) 2000-2500 m, (f) above 2500 m.

reduced, meaning that less snow is added at higher elevations in $DA_{var}$. While the OL has a negative snow bias in these higher elevation areas, the weaker positive increments in $DA_{var}$ may be more realistic, given that Figure 4b indicates a strong positive snow depth bias for sites about 2500 m in $DA_{const}$, and a reduced positive bias at these same sites in $DA_{var}$.

The change in observation uncertainty also has an impact on the analysis ensemble spread, with primarily decreased ensemble spread in $DA_{var}$, compared to $DA_{const}$, especially in lower elevation regions (Supplemental Fig. S7a). Changes in analysis spread are related to changes in the observation uncertainty, with decreases in spread corresponding to decreases in average observation uncertainty (Supplemental Fig. S7b). For example, for all model grid cells where $\sigma_{obs}$ decreases, on average, from $DA_{const}$ to $DA_{var}$, 83% indicate a corresponding decrease in the snow depth analysis ensemble spread. In contrast, for all grid cells where $\sigma_{obs}$ increases, 65% have a corresponding increase in analysis ensemble spread. The reason for this decrease in ensemble spread is likely two-fold. This overall decrease in ensemble spread is likely driven by two factors: (1) lower observation uncertainty in many regions, and (2) reduced snow depth, which results in smaller multiplicative perturbations to the forecast state.

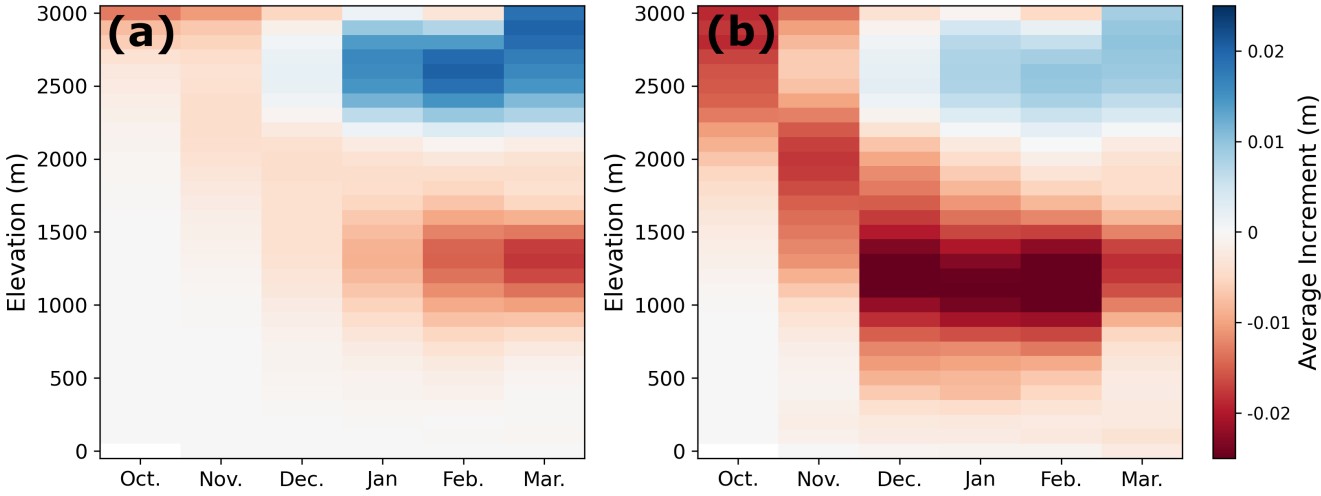

**Figure 9.** Average total snow depth increments (m) over the accumulation season (x-axis), stratified by elevation (y-axis) for (a) $\text{DA}_{\text{const}}$ and (b) $\text{DA}_{\text{var}}$. Increments are averaged over all years (2015-2023)

## 4 Discussion

This work enhances snow DA by incorporating an ML-based snow depth retrieval product using spatio-temporally dynamic error estimates into the assimilation scheme. The ML snow depth retrieval integrates multiple sources of information, including S1 backscatter observations, fractional snow cover from optical imagery, and land cover information to estimate snow depth. Future work could experiment with integrating additional satellite-based information into the assimilated ML product (e.g., passive microwave, X-band, lidar data). The snow depth estimated from this ML model has been shown to possess superior accuracy compared to prior S1 snow depth retrieval work by Lievens et al. (2022) ($SD_{S1}$) (Dunmire et al., 2024), which has previously been assimilated into the Noah-MP land surface model using an Ensemble Kalman Filter (De Lannoy et al., 2024; Brangers et al., 2024). Recent work by Mirza et al. (2025) has questioned the utility of assimilating S1 snow depth retrievals, highlighting inconsistencies in temporal and spatial errors of the $SD_{S1}$ in the Western United States, where less regular S1 data are available. Despite advancements made by $SD_{ML}$, the quality of the ML-based observations assimilated in this study also varies across space and time, which can lead to localized degradations in DA performance (e.g., Fig. 3). Although improving mountain snow-depth estimation is an active area of research, progress is limited by the current suite of satellite sensors, which are not specifically designed for snow-depth or SWE retrieval. Future DA efforts that incorporate more reliable snow-depth or SWE products should reduce these spatial and temporal inconsistencies, improving overall DA performance.

In the OL, we see an overestimation of SWE at measurement sites with low recorded SWE, and an underestimation of SWE at measurement sites with high recorded SWE (Fig. 5c). Previous work has demonstrated that forcing bias is the dominant source of uncertainty in snow modeling (Raleigh et al., 2015). Here, we use ERA5 atmospheric forcing, which has a relatively coarse spatial resolution (31 km). While we apply a standard lapse-rate correction to downscale the near-surface air temperature

forcing, precipitation is not downscaled, and therefore is unable to resolve orographic precipitation. This limitation results in

relatively low precipitation and SWE spatial variability and an underestimation of high SWE values. Furthermore, the $SD_{ML}$ product has also been demonstrated to underestimate deep snow, likely due to these measurements being underrepresented in the ML training (Dunmire et al., 2024). As such, the assimilation of this product is unable to fully correct the negative SWE bias for measured SWE $> \sim 800$ mm, as can be seen in Figure 5d/e.

Here, we also highlight the implications of accounting for dynamic estimates of the observation uncertainty and demonstrate

that this system generally results in a more realistic modeled snow state. The EnKF depends on accurate uncertainty estimates for both the model and observations, using these to weigh the information and obtain an optimal state. With this in mind, Dee (1995) argues that proper characterization of both model and observation uncertainties is necessary for successful implementation of the EnKF. While the specification of observation uncertainty influences DA performance, in snow DA systems, this uncertainty is often prescribed as a constant value (Helmert et al., 2018). Some previous studies have incorporated dynamic

observations errors (e.g., Magnusson et al. (2017); Oberrauch et al. (2024)); however, the utility of dynamic observation errors, relative to an assumed static observation error, in snow DA has not yet been explored prior to this work. Moreover, most operational land data assimilation systems (e.g., NASA Land Data Assimilation Systems, ECMWF Land Data Assimilation System) and recent studies that assimilate SAR-based snow depth retrievals assume a static observation error. For instance, Brangers et al. (2024) assumed $\sigma_{obs} = 0.36$ $m$, and Girotto et al. (2024)) and De Lannoy et al. (2024) both assume $\sigma_{obs} = 0.30$ $m$

(applied here in $DA_{const}$).

We find that assimilating $SD_{ML}$ with a dynamic observation error ($DA_{var}$) offers a significant improvement to SWE MAE ($p \ll 0.001$) compared to assimilating $SD_{S1}$ with a static observation error ($DA_{S1}$, Supplemental Fig. S8). Meanwhile, $DA_{const}$ does not demonstrate any significant improvements to SWE MAE (Supplemental Fig. S8). Using 4548 manual SWE measurements collected within the Po River basin, we find an MAE of 225 mm from the OL experiment, while the MAE for

$DA_{S1}$, $DA_{const}$, and $DA_{var}$ is 193, 195, and 177 mm, respectively. Generally, the $SD_{ML}$ retrievals are more accurate than $SD_{S1}$ for in-situ snow depths below 2.5 m, while for snow depth exceeding 3 m, $SD_{S1}$ performs better (see Figure 3a from Dunmire et al. (2024)). This suggests that assimilating $SD_{ML}$ should provide improvements particularly for shallower snow. However, in the $DA_{const}$ experiment, the use of a static observation uncertainty, where relatively large errors are assumed for shallow snow observations, limits these potential improvements (e.g. Fig. 2a) and results in an overall performance of $DA_{const}$

that is similar to $DA_{S1}$. This analysis highlights that the treatment of the observation uncertainty can be as critical as the observations themselves. A poorly parameterized observation uncertainty can restrict the benefits of DA, underscoring the need for options in DA systems to dynamically vary the observation error.

Implementing the dynamic observation error generally improves performance in both places the DA adds and removes snow. In the OL experiment, snow depth has a positive bias at low elevations and a negative bias at high elevations (Fig. 4a). The

$DA_{const}$ experiment applies a static observation error that is relatively too large for shallow assimilated snow depths (e.g. Fig. 2a), limiting snow removal at lower elevations and leading to a still large positive bias at these locations. At higher elevations (above $\sim 1500$ m), the assimilated observations exhibit a strong positive bias (Fig. 4d). The relatively small static observation error for deeper assimilated snow depths (e.g. Fig. 2b) leads to too much added snow in some cases, particularly above 2500 m

(Fig. 4b). In contrast, in $\mathrm{DA_{var}}$, less snow is added at high elevations (Fig. 9), resulting in improvements where snow needs
to be added as well (Fig. 4f). However, we see that the $\mathrm{DA_{var}}$ experiment performs worse than $\mathrm{DA_{const}}$ between February
and May within the 2000-2500 m elevation band (Fig. 4f). In this range, the OL experiment has a positive snow depth bias
until approximately February, followed by a negative snow depth bias until May (Fig. 4a). $\mathrm{DA_{var}}$ more effectively reduces
this early season positive bias, resulting in lower mean snow depths later in the season, and poorer performance during the
period when the OL is negatively biased. This suggests that a lack of early-season corrections in $\mathrm{DA_{const}}$ can, in some cases,
propagate to more accurate late-season snow depths, although this effect is likely limited to locations where the snow depth is
not consistently positively or negatively biased throughout the season.

While $\mathrm{DA_{var}}$ improves performance at most snow depth and SWE measurement sites, some locations see little benefit, or
even a deterioration in performance (approximately 12% of snow depth sites and 20% of SWE sites). These degradations
are more likely to occur where the $SD_{ML}$ product is less accurate than the OL experiment, and the $DA_{var}$ experiment more
strongly corrects to these inaccurate observations. To account for known limitations of SAR-based snow depth retrievals, we did
not assimilate the $SD_{ML}$ product over dense forests or glaciers, and after March 31. Nevertheless, $SD_{ML}$ remains inaccurate
in some places, leading to localized deterioration when these observations are assimilated. Locations with minimal differences
between $\mathrm{DA_{const}}$ and $\mathrm{DA_{var}}$ typically occur where the observations already agree well with the OL, or where $\sigma_{obs} >> \sigma_f$,
thus the DA increments are small, and the model receives limited benefit from the observational information. Despite these
spatial inconsistencies, $\mathrm{DA_{var}}$ nearly doubles the improvement in absolute SWE error compared to $\mathrm{DA_{const}}$. For instance, the
SWE MAE decreases from 152 mm in $\mathrm{DA_{const}}$ to 132 mm in $\mathrm{DA_{var}}$ (-13.2%), while the overall impact of $\mathrm{DA_{const}}$ relative to
the OL is a 13.6% reduction (176 mm in the OL to 152 mm in $\mathrm{DA_{const}}$).

Snow cover fraction affects the energy balance, and consequently, has implications for numerical weather prediction. While
the DA experiments generally reduce the snow-covered area by largely removing snow at lower elevation regions, all three
experiments still exhibit an overestimation of total snow-covered area compared with both Copernicus and IMS snow cover
products. Several factors may contribute to this discrepancy. First, a positive bias in snowfall forcing data at low elevations will
result in unrealistically large snow-covered area. Second, the higher-resolution Copernicus product (20 m) inherently captures
finer-scale variation between snow-covered and snow-free conditions, often resulting in lower overall snow cover estimates
compared to coarser-resolution products. Third, inaccuracies in the parameterization of snow cover fraction within Noah-MP
may also play a role. In Noah-MP, the snow cover fraction is parameterized as a function of snow depth, density, and ground
roughness length. (Niu et al., 2011; Lee et al., 2024). It should be investigated whether the current parameterizations in Noah-
MP remain appropriate for regions with complex terrain, where subgrid variation in topography can influence fractional snow
cover. Finally, uncertainty in the Copernicus and IMS snow cover, for example due to cloud and forest cover, contribute to
errors in these validation data sets and potentially influence the perceived model biases.

Finally, in $\mathrm{DA_{var}}$, Equation 2 ($\sigma_{obs} = m * SD_{ML}$, $m = 0.3$) is used to adapt the standard deviation of the observation
error in space and time based on the assimilated snow depth. This relationship is a first-order approximation that assumes
that the observation error increases linearly with the observation magnitude; however, $\sigma_{obs}$ could be defined to vary in more
complex ways. Future work could explore applying relationships where $\sigma_{obs}$ varies non-linearly with the assimilated snow

depth observation, or statistical parameterizations of $\sigma_{obs}$ depending on other conditions such as elevation, or forest cover.
Furthermore, $\sigma_{obs}$ could be directly linked to the $SD_{ML}$ retrieval quality which could be obtained, for example, through error propagation. The effectiveness of a dynamic observation error also depends on the magnitude of the forecast error, as the Kalman gain matrix, which determines the strength of the corrections, depends on both forecast and observation error. To maximize benefits, the observation error, whether static or dynamic, should be properly tuned in relation to forecast error. While most operational systems do not currently include options to dynamically vary the observation error, this functionality is
not complicated to incorporate, and the snow-specific MuSA (Multiple Snow Data Assimilation System) system does already provide an option for a user-defined observation error that can vary dynamically (Alonso-González et al., 2022).

## 4.1   Limitations of bias-blind DA systems

The EnKF is widely used in snow DA systems due to its efficiency; however, a key assumption is that both the observations and model are unbiased. We see from Figure 4 that this assumption is not satisfied by neither the observations nor the model.
Here, we implement a bias-blind system by not bias-correcting either the observations or the model, thereby violating this assumption. Bias-aware systems which a priori correct the model bias to align with the observation climatology assume that the assimilated observations are more realistic than the model. While this assumption may be realistic in many situations, satellite-based snow retrievals also have inherent biases. Since snow is a cumulative variable, biases in either the observations or the model typically persist throughout the snow season. While in-situ measurement stations can help quantify these biases,
they are often inconsistent spatially and on an interannual basis (i.e. Supplemental Fig. S9), which provides a challenge for correcting them a priori.

Two major issues exist with bias-blind systems: (1) model drift towards its original state, leading to a sawtooth-like pattern that can result in unrealistic fluxes in other variables, and (2) unrealistic model trends in DA output due to changes in assimilated observation frequency (Dee, 2005). For snow, model biases primarily stem from errors in precipitation forcing data.
Consequently, we do not expect model drift to occur as observed in De Lannoy et al. (2007); Mocko et al. (2021); Scherrer et al. (2023), unless there is an instantaneous precipitation forcing error. We also assimilate observations weekly throughout the study period, thereby mitigating the potential effects of assimilation frequency in bias-blind DA. Scherrer et al. (2023) further compare bias-blind and bias-aware assimilation of leaf area index - a cumulative variable - using the EnKF. Their results show that the bias-blind DA more effectively updates the model state variable, and leads to larger improvements in water balance
components such as evapotranspiration and runoff. In contrast, while the bias-aware approach yields smaller improvements in state variables, it improves temporal anomalies and internal DA diagnostics indicate a more optimal DA system performance. Given our focus on improving the modeled snow state rather than snow anomalies, along with the inherent challenges of a priori bias correcting the observations and model, we opt for a bias-blind approach, recognizing that this may lead to suboptimal DA performance (i.e. temporally correlated residuals).

## 4.2 Limitations of site evaluation representativeness

Previous studies have shown that mountain snow is highly variable, and point-scale measurements don't necessarily well-represent the surrounding area, even at spatial scales as fine as 10 m (López-Moreno et al., 2011; Fassnacht et al., 2018). Meromy et al. (2013) found that approximately half of the SNOTEL sites they analyzed where representative of the surrounding 1 km area, defining "representative" as snow station biases within 10% of the surrounding mean observed depth. More recently, Herbert et al. (2024) reported that roughly one-third of 476 paired lidar–station data observations were representative at the 1 km scale, with representativeness defined as in-situ measured snow within ±10 cm of the lidar-mean snow depth at that scale. However, they also showed little change between the 500 m and 1 km scales, with 35% of stations considered as representative at 500 m. Generally, in-situ snow stations exhibit a positive bias as these sites are often located in flat terrain that preferentially accumulates snow (Grünewald and Lehning, 2011).

In this study, we use in-situ snow depth and manual SWE measurements as the best-available reference in the European Alps that cover a range of terrain conditions and spans many years. Unlike in the western United States, where high-resolution spatial snow depth products from the Airborne Snow Observatory and NASA SnowEx missions are available, such publicly available products are extremely limited in the European Alps. As such, it is not feasible to assess the representativeness of all 588 snow depth measurement sites and 8211 manual SWE measurements at the 1 km scale, and these point-scale measurements provide the best available Alps-wide, multi-year data available for evaluation. Nevertheless, by leveraging a large network of sites that span a range of elevations and terrain types, we can reduce sampling-related limitations by increased coverage of terrain diversity, although this does not mitigate the general positive bias noted above.

## 5 Conclusions

In this work, we explore how incorporating a dynamic observation uncertainty can influence a snow depth data assimilation scheme. For the first time, we assimilate satellite-based snow depth estimates from a novel machine learning model into the Noah-MP land surface model using the EnKF to update snow depth and SWE. We compare two data assimilation experiments: one with a static observation error ($DA_{const}$), and one with an observation error that is dynamic in space and time ($DA_{var}$). The performance of these DA experiments is evaluated against the open-loop experiment (OL, model-only) using in-situ snow depth observations, manual SWE measurements, and two different snow cover products. Overall, the dynamic observation error appears to make better use of the assimilated observations, thereby leading to stronger model corrections, particularly at times when the assimilated snow depth observation is much shallower than the model forecast (e.g., early in the accumulation period or at lower elevations). By doing so, $DA_{var}$ reduces biases tied to forcing errors and improves SWE MAE by 25% and 13% compared to the OL and $DA_{const}$ experiments, respectively. While snow cover is overestimated in all three model experiments, $DA_{var}$ also leads to stronger reductions in snow cover than $DA_{const}$, better aligning with existing snow cover products. However, given limitations of the assimilated satellite-based snow depth product, improvements from the DA, or from the specific implementation of a dynamic observation error in $DA_{var}$, are limited in magnitude and not spatially consistent. As most snow DA work and operational snow DA systems assume that the observational uncertainty is constant in space

and time, this work highlights the impact of these assumptions, and the importance of observation uncertainty considerations when designing a DA system. Future studies should put effort into the consideration of observation uncertainties and the parameterization of observation uncertainty should depend on study goals, the DA system used, and specific characteristics of the assimilated observations.

*Code and data availability.* The ML-based snow depth retrieval product is publicly available at https://doi.org/10.5281/zenodo.13342108 (Dunmire, 2024). The NASA LIS software is available at https://github.com/NASA-LIS/LISF. The IMS snow cover dataset is publicly available at: https://nsidc.org/data/g02156/versions/1anchor-data-access-tools (U.S. National Ice Center, 2008), and the Copernicus snow cover product can be found at: https://land.copernicus.eu/en/products/snow/fractional-snow-cover (European Union's Copernicus Land Monitoring Service information). Publicly available in-situ snow depth and SWE data used for evaluation can be accessed at:

- https://www.doi.org/10.16904/15 (Switzerland; Marty (2020))
- https://www.doi.org/10.16904/envidat.380 (Switzerland; Stähli (2018))
- https://www.doi.org/10.16904/envidat.590 (Switzerland; Magnusson and Jonas (2025))
- https://www.doi.org/10.16904/envidat.406 (Switzerland; Intercantonal Measurement and Information System IMIS (2023))
- https://www.arpa.piemonte.it/rischi_naturali/snippets_arpa_graphs/map_meteoweb/?rete=stazione_meteorologica (Italy)
- https://www.meteotrentino.it/index.html#!/home (Italy)
- https://data.civis.bz.it/de/dataset/p-bz-southtyrolean-weatherservice-weatherstations/resource/ef2f6f24-cffd-4993-8699-5023696a49b5 (Italy)
- https://dataset.api.hub.geosphere.at/app/frontend/station/historical/klima-v2-1d (Austria)
- https://donneespubliques.meteofrance.fr/?fond=recherche (France)
- https://cdc.dwd.de/portal/ (Germany)

Additional snow depth and SWE data were obtained from the Italian Department of Civil Protection and processed by the Centro Internazionale in Monitoraggio Ambientale (CIMA).

The configuration files used for the modeling experiments can be found at https://github.com/drdunmire1417/Snow_DA_LIS_configfiles.git.

*Author contributions.* DD and GDL conceived and designed the study. DD conducted the model experiments, with help from MB. Material preparation and data collection were performed by DD and LB. Formal analysis, visualization, and original draft preparation was done by DD and all authors contributed to the manuscript reviewing and editing.

*Competing interests.* The authors declare that there are no competing interests.

*Acknowledgements.* This work was funded by the project C14/21/057 of KU Leuven and SNOWTRANE (SR/00/407) of the Belgian Science Policy (Belspo). The computer resources and services were provided by the High Performance Computing system of the Vlaams Supercomputer Center, funded by FWO and the Flemish Government (incl. Storage4Climate collaborative grant). The authors acknowledge CIMA for providing reference data. During the preparation of this work the authors used ChatGPT in order to improve readability, flow, and language.
After using this tool, the authors reviewed and edited the content as needed and take full responsibility for the content of the publication.

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
