# Peer review of "Advancing snow data assimilation with a dynamic observation uncertainty"

_EGUsphere, 2025_

## Referee Comment (RC1)

**Overall Comment:**

This manuscript explores the assimilation of a machine learning-derived Sentinel-1 snow depth product into a NOAH land surface model using a dynamically varying observation error. While the approach is innovative and addresses an important limitation in current snow data assimilation systems, the demonstrated improvements over the static error method are relatively modest/marginal and not consistent across space or time. Given the added complexity of implementing a dynamic error model, I do not fully agree with the authors' conclusion that this approach provides a clear performance advantage. The manuscript has potential, particularly if it reframes the findings to emphasize that the benefits of dynamic error observation methods are highly dependent on the pattern and variability of observation errors. One has minimal leverage on the other, and a more thorough error characterization is essential for selecting an appropriate DA strategy for the problem at hand.

**Minor Comments:**

**Comment 1- Introduction:**

Since this paper focuses on evaluating different data assimilation (DA) approaches, it would strengthen the manuscript to include a more thorough overview of existing DA algorithm literature, for context and completeness. I recommend adding this near Lines 44–51, where the background on DA methods is introduced.

**Comment 2 – Methods/Results:**

Justify the use of 0.05 to 1.05 bounds and increasing error with time. It would be helpful to include spatial and temporal figures (e.g., from representative sites) comparing the input observations against independent validation data. This could illustrate the spatial and temporal variability of observation errors and help justify the bounds selected for dynamic error modeling (0.05 to 1.05). For instance, do errors vary systematically with snow depth, such as being lower in shallow snow and higher in deeper snow? As noted by Alonso-González et al. (2024), no data assimilation algorithm is universally superior; performance depends on the data and task at hand. Thus, this

figure can further support the choice of the Ensemble Kalman Filter (EnKF) and observation error technique.

**Major Comments:**

**Comment 1 - Result:**

While the authors claim that the Dvar experiment outperforms DAconst in terms of snow depth and SWE estimation, the practical improvement is marginal at best. For snow depth, the spatial ACC increases only slightly—from 0.72 (DAconst) to 0.73 (DAvar)—a mere 0.01 gain (Figure 3c), which is unlikely to represent a meaningful enhancement in most applications. The temporal ACC comparison (Figure 3d) shows that just ~52% of sites improve with DAvar while nearly half do not, and 11% of sites degrade by more than 0.02. Similarly, the MAE reduction from DAconst to DAvar occurs at only ~51% of sites, with 12% worsening. These statistics reveal that the advantage of DAvar is not robust or generalizable. The pattern is echoed in the SWE evaluation, where only 56% of sites see improvement in MAE under DAvar compared to DAconst. In other words, nearly half the sites experience no benefit or deterioration, which raises questions about the reliability of the variable uncertainty approach. This is consistent with SDD and SCF evaluation (Figures 7 and 8). It is also surprising that RMSE was not reported, as it is a standard metric in snow modeling and data assimilation evaluations, and helps better in error magnitude in snow depth and SWE.

Despite the narrative of statistical significance (e.g., $p < 0.001$), these results suggest that the magnitude of improvement is small, the spatial consistency is weak, and the operational gain may not justify the added complexity. The authors should contextualize these findings more carefully, perhaps by comparing the computational cost or exploring why improvements are minimal across the full domain.

**Comment 2 - Discussion:**

The discussion attempts to cover many important aspects of the study. However, it overemphasizes statistical significance while underplaying the marginal and spatially inconsistent nature of the improvements. Standard metrics like RMSE are missing to give a clear

picture. Key ideas like bias treatment and dynamic observation error comparison with SDs1 are introduced without prior mention or clear connection to the results or existing literature.

**Line to Line Comments:**

**Lines 17–29** would be more effective as a single cohesive paragraph. The current break into two paragraphs disrupts the logical flow and makes the message harder to follow.

**Lines 30-43** discuss various SWE estimation methods, but some other approaches such as spaceborne laser altimetry (e.g., ICESat/ICESat-2), Sentinel-2, and MODIS, are missing and should be acknowledged for completeness. Additionally, the paragraph uses SWE and snow depth somewhat interchangeably (line 42). It would strengthen the clarity to note explicitly that snow density is required to convert snow depth into SWE.

**Line 58-59:** particle batch filters and smoothers.. can be more computationally expensive given the large number of particles required." While this is generally true for particle filters, it is not necessarily the case for particle batch smoothers (PBS). For example, Alonso-González et al. (2024) showed that PBS was less computationally expensive than EnKF and other particle-based methods across multiple particle counts (100, 200, 300). The authors should either revise the statement to reflect this variability or cite relevant studies to justify the statement.

**Line 67-86:** Report the accuracy metrics from Lievens et al. compared to those from Dunmire et al. (2024). Additionally, please clarify that higher accuracy was achieved when evaluated over the European Alps only, as this distinction is crucial for understanding the geographic limitations of these methods.

**Line 62-73:** Clarify that the primary goal of this work is to assess the utility of incorporating dynamic observation errors versus static ones, because there are studies that have compared the performance of different algorithms already.

**Line 101-102:** specify the reported accuracy metrics (same as 67 and 86). What does better mean?

**Line 173-184**: The evaluation section introduces Snow Disappearance Date (SDD) and Snow Cover Fraction (SCF) without prior mention or justification in the earlier sections of the

manuscript. To improve clarity and coherence, it would be helpful to introduce these variables earlier in the manuscript, explain their relevance to the study objectives.

**Figure 2:**

- Why were these two sites chosen? State reason (representativeness, topography, etc) either in the figure description or the result section.
- Axis can be shared for better readability (general comment for all figures).

Line 240-243: State the bias-corrected numbers for the constant as well and compare them to the variable. "Both DAconst and DAvar also substantially improve the Pearson correlation coefficient…" report numbers.

**Line 325:** The claim that dynamic observation error estimation "had not yet been explored" overlooks prior work that has implemented such approaches (e.g., Alonso-González et al., 2022). While these studies may not compare dynamic and constant errors directly, they do demonstrate prior use of dynamic error treatment in snow science. I suggest rephrasing to acknowledge existing efforts and clarify this study's specific novelty.

---

## Author Comment (AC2)

**Reviewer 2**

This manuscript performs an assimilation experiment over the European Alps to investigate several advances to the modeling and assimilation scheme. Specifically, they consider the assimilation of a snow depth machine learning product based on Sentinel-1 observations and the development of a dynamic observation error for their Ensemble Kalman Filter setup. I find this study to be very compelling with interesting results, and I only have a few minor comments.

We thank this reviewer for their thoughtful and helpful comments and we believe that our revised manuscript is much improved as a result. Our responses below, are in red, with manuscript text in dark red and *additions to the manuscript in italics.*

1) Are the snow depth ML estimates available everywhere and thus assimilated everywhere or are there flags to exclude assimilation in higher uncertainty areas where SAR does not perform well, like dense forests?

The final sentence in **Section 2.3 Data assimilation approach and experiments** explains the flags used to exclude assimilation under certain conditions. We will modify this sentence slightly so that these conditions are more obvious:

"We assimilated $SD_{ML}$ estimates weekly each year from September 1 through March 31, *excluding assimilation further into the ablation period when wet snow complicates the S1 signal. Due to limitations of S1 in forested terrain,* and the unsuitability of the ML SD retrieval over glaciated terrain, we do not assimilate over forested areas or glaciers. Following De Lannoy et al. [2024], we also do not assimilate when the soil or vegetation temperature is above 5 °C."

2) How representative are the in situ observations of the surrounding ~1km to match the modeled grid?

This is a good question. To address this, we will add the following section to the Discussion:

**4.2 Limitations of site evaluation representativeness**

*Previous studies have shown that mountain snow is highly variable, and point-scale measurements don't necessarily well-represent the surrounding area, even at spatial scales as fine as 10 m [López-Moreno et al., 2011, Fassnacht et al., 2018]. Meromy et al. [2013] found that approximately half of the SNOTEL sites they analyzed where representative of the surrounding 1 km area, defining "representative" as snow station biases within 10% of the surrounding mean observed depth. More recently, Herbert et al. [2024] reported that roughly one-third of 476 paired lidar–station data observations were representative at the 1 km scale, with representativeness defined as in-situ measured snow within $\pm 10$ cm of the lidar-mean snow depth at that scale. However, they also showed little change between the 500 m and 1 km scales, with 35% of stations considered as representative at 500 m. Generally, in-situ snow stations*

*exhibit a high bias as these sites are often located in flat terrain that preferentially accumulates snow [Grünewald and Lehning, 2011].*

In this study, we use in-situ snow depth and manual SWE measurements as the best-available reference in the European Alps that cover a range of terrain conditions and spans many years. Unlike in the western United States, where high-resolution spatial snow depth products from the Airborne Snow Observatory and NASA SnowEx missions are available, such products are extremely limited in the European Alps. As such, it is not feasible to assess the representativeness of all 588 snow depth measurement sites and 8211 manual SWE measurements at the 1 km scale, and these point-scale measurements provide the best available Alps-wide, multi-year data available for evaluation. Nevertheless, by leveraging a large network of sites that span a range of elevations and terrain types, we can mitigate some of the inherent limitations of using point measurements for evaluating the 1 km gridded product.

3) Are there any spatial datasets, like lidar, for evaluating modeled snow depth over a broader area?

Unfortunately, spatial datasets such as lidar are very limited over the European Alps. A number of airborne photogrammetry products exist for the Dischma Valley in Switzerland, but comparable, freely available datasets are non-existent elsewhere in the Alps. Moreover, most of these available spatial snow depth datasets (with the exception of two products analyzed in Dunmire et al. [2024]) were used in training the $SD_{ML}$ product and therefore cannot be considered as independent evaluation sources. Given the extremely limited spatial and temporal coverage of these spatial datasets, we instead focus our evaluation on in-situ snow depth stations and manual SWE measurements. These datasets, collected across multiple nations, span a range of elevations and terrain conditions, and provide continuous multi-year records, offering a more robust basis for assessing the impact of the DA.

4) In Figure 2b, the OL performs better than either DA scenario compared to the in situ, though $DA_{var}$ performs better than $DA_{const}$. Is this similar behavior in other locations where the initial OL estimate is already close to the in situ truth?

Yes, inevitably there are locations where the OL model run is closer to reality than the observations and the assimilation leads to a deterioration. Figure 3a demonstrates how frequently this phenomenon occurs. We can see here that DA leads to a deterioration in MAE at 16% of Alps-wide measurement sites, with only ∼1% experiencing a deterioration > 125 mm. We will include this information in the text of the manuscript as follows:

"A measurement site with assimilated snow depths substantially greater than 1 m is demonstrated in Figure 2b. In this case, the observation uncertainty is smaller for $DA_{const}$ than for $DA_{var}$, resulting in stronger posterior state adjustments in $DA_{const}$. *At this measurement site we see that the OL experiment is closer to the in-situ snow depth than the assimilated observations, leading to a deterioration in model performance when the DA is applied. This phenomenon occurs at 16% of all measurement site (Fig. 3a), with only 1% experiencing a deterioration in SD MAE greater than 125 mm.*"

5) How does the DA perform when snow needs to be added to the system? Figure 2a shows the DA reducing the modeled snow down to match the observations. Though in later results, it appears that DA mostly reduces snow in the model.

Here the DA does mostly reduce snow because the OL has a positive snow depth bias due to an overestimation of precipitation at lower elevations in ERA5. However, we can see in Figure 9 that both $\mathrm{DA_{const}}$ and $\mathrm{DA_{var}}$ generally add snow above approximately 2250 m. We will add the following paragraph in the Discussion section to discuss DA performance when snow needs to be added to the system:

"Here, we highlight the implications of accounting for dynamic estimates of observation uncertainty and demonstrate that this system results in a more realistic modeled snow state. *Implementing the dynamic observation error generally improves performance in both places the DA adds and removes snow. In the OL experiment, snow depth has a positive bias at low elevations and a negative bias at high elevations (Fig. 4a). The* $\mathrm{DA_{const}}$ *experiment applies a static observation error that is relatively too large for shallow assimilated snow depths (e.g. Fig. 2a), limiting snow removal at lower elevations and leading to a still large positive bias at these locations. At higher elevations (above ~1500 m), the assimilated observations exhibit a strong positive bias (Fig. 4d). The relatively small static observation error for deeper assimilated snow depths (e.g. Fig. 2b) leads to too much added snow in some cases, particularly above 2500 m (Fig. 4b). In contrast, in* $\mathrm{DA_{var}}$*, less snow is added at high elevations (Fig. 9), resulting in improvements where snow needs to be added as well (Fig. 4f).*"

We will also add a new panel to Figure 4 (see Review Figure 1), showing the bias of the assimilated observations for the various elevation bands.

6) Lines 226-230: You mentioned that some of the increases in MAE at higher elevations could be due to limitations in the model during the melt season or biases from the assimilation. In your results, do you find that DA tends to add to remove snow at the higher elevations? How does the ML snow depth product perform at higher elevations where I assume the snow is deeper.

We can see from Figure 9 that both $\mathrm{DA_{const}}$ and $\mathrm{DA_{var}}$ both add snow above approximately 2250 m. To address this comment, we will add a subpanel, depicting the bias of the $SD_{ML}$ assimilated observations for different elevation bands, to Figure 4 (See Review Figure 1). This panel indicates that the assimilated observations exhibit minimal bias below 1500 m and a positive bias above 1500 m. We will modify Lines 226-230 as follows: "However, an increase in MAE at high elevations during the melt season (March onwards) suggests a tendency for the DA experiments to retain snow for too long, which could be due to limitations in the modeled melt processes or biases introduced by the assimilated observations at higher elevations *(e.g., Figure 4d)*."

7) Figure 4e: Do you have an idea why $\mathrm{DA_{var}}$ degrades performance around peak accumulation for the 2000-2500 m elevation band? Aside from that line, it appears like $\mathrm{DA_{var}}$ improves upon $\mathrm{DA_{const}}$ for each elevation band throughout the winter

[Figure]

Review Figure 1: (Revised Figure 4 in manuscript) Seasonal evolution of bias and mean absolute error (MAE) stratified by elevation. Panels (a)-(d) show the seasonal snow depth bias for the (a) OL, (b) $DA_{const}$, and (c) $DA_{var}$ experiments, and for *(d) the assimilated observations ($SD_{ML}$).* Bias is computed relative to in-situ snow depth measurements and is grouped by elevation bands (indicated by different colors). Panels (e)–(f) show the change in MAE between the OL and $DA_{var}$ experiments (e) and between the $DA_{const}$ and $DA_{var}$ experiments (f). Negative values in (e)–(f) indicate improved performance (decreased MAE). Statistics are computed for each day, averaged over the entire 8-year period (2015–2023). A 14-day smoothing is applied to each timeseries and the number of in-situ measurement sites within each elevation band is provided in the legend.

[Figure]

Review Figure 2: Seasonal evolution of snow and bias for sites in the 2000-2500 m elevation band. (a) Seasonal snow depth bias for the OL (blue) and the assimilated observations (black). Bias is computed relative to in-situ snow depth measurements from sites within this elevation band. (b) Mean snow depth at these same in-situ measurement sites from $DA_{const}$ (red) and $DA_{var}$ (orange).

Yes, Review Figure 2 shows side-by-side comparison timeseries of the bias for the OL and the assimilated observations for sites between 2000 and 2500 m elevation. We see that the OL has a positive snow depth bias until approximately February and then a negative snow depth bias until May, while the assimilated observations have a positive bias throughout the entire snow season. However, in the early season (before January), the assimilated observations have a lower magnitude bias than the OL experiment. As discussed throughout the manuscript, $DA_{var}$ leads to more effective snow reductions due to its lower assumed observational uncertainty for shallow assimilated snow depths. As such, in February, $DA_{var}$ has a lower average snow depth than $DA_{const}$ for sites between 2000 and 2500 m (Review Figure 2). This indicates that a lack of early season corrections in $DA_{const}$ can propagate to better simulated late-season snow, although this impact is limited to specific to sites where the snow depth bias is not consistent throughout the snow season.

We will discuss this phenomenon in the following text in the Discussion section:

*"However, we see that the $DA_{var}$ experiment performs worse than $DA_{const}$ between February and May within the 2000-2500 m elevation band (Fig. 4f). In this range, the OL experiment has a positive snow depth bias until approximately February, followed by a negative snow depth bias until May (Fig. 4a). $DA_{var}$ more effectively reduces this early season positive bias, resulting in lower mean snow depths later in the season, and poorer performance during the period when the OL is negatively biased. This suggests that a lack of early-season corrections in $DA_{const}$ can, in some cases, propagate to more accurate late-season snow depths, although this effect is likely limited to locations where the snow depth is not consistently positively or negatively biased throughout the season."*

8) Figure 5: It appears like the modeled SWE almost reaches an asymptote around 600-700mm of SWE. Any ideas why the model is underestimating at the deepest SWE values?

This apparent asymptote is likely primarily due to limits with both the precipitation forcing and the observations. Indeed, Raleigh et al. [2015] demonstrate that forcing bias is the dominating uncertainty source in snow modeling. The atmospheric forcing used in this study comes from the relatively low resolution ERA5 atmospheric reanalysis product (31 km horizontal resolution). As such, the precipitation forcing for the land surface model has a coarse horizontal resolution and is unable to resolve orographic precipitation, resulting in an underestimation of precipitation at high elevations and a corresponding underestimation at high SWE. From Figure 2b in Dunmire et al. [2024] we see a similar asymptotic behavior of $SD_{ML}$ for recorded snow depths above ∼3.5 m, likely attributed to these deep snow measurements being underrepresented in the ML training. To discuss this in the manuscript we will add the following paragraph in the Discussion section

*"In the OL, we see an overestimation of SWE at measurement sites with low recorded SWE, and an underestimation of SWE at measurement sites with high recorded SWE (Fig. 5c). Previous work has demonstrated that forcing bias is the dominant source of uncertainty in snow modeling [Raleigh et al., 2015]. Here, we use ERA5 atmospheric forcing, which has a relatively coarse spatial resolution (31 km). While we apply a standard lapse-rate correction to downscale the near-surface air temperature forcing, precipitation is not downscaled, and therefore is unable to resolve orographic precipitation, resulting in relatively low precipitation and SWE spatial variability, and an underestimation of high SWE values. Furthermore, the $SD_{ML}$ product has also been demonstrated to underestimate deep snow, likely due to these measurements being underrepresented in the ML training [Dunmire et al., 2024]. As such, the assimilation of this product is unable to fully correct the negative SWE bias for measured SWE > ∼800 mm, as can be seen in Figure 5d/e."*

9) Figure 7: The difference plot for $DA_{var}$ is much jumpier than that from $DA_{const}$. Can you explain that? Does $DA_{var}$ have more ephemeral snow that comes and goes throughout the winter and spring? I assume much of that is from low elevation snow?

Yes, in $DA_{var}$ snow at low elevation and early in the snow season (October, November) is often lower than in $DA_{const}$ and therefore melts away more quickly, resulting in the jumpier nature of the $DA_{var}$ difference timeseries. We will add the following sentence to the manuscript to describe this phenomenon: *"The relative difference in snow-covered area between $DA_{var}$ and the OL fluctuates more than for $DA_{const}$ (Fig. 7a), primarily due to the shallower early-season and low-elevation snowpacks in $DA_{var}$ which melt out more quickly."*

10) Lines 264-268: Do you have any time series with snow cover comparisons to IMS or Copernicus?

It would be difficult to compare snow cover time series with the Copernicus product. The Copernicus Fractional snow cover product is not gap filled, meaning that the data is not spatiotemporally continuous. The model's strong overestimation of snow cover (described in Section 3.3, and in L360-371) compared to the IMS product is temporally consistent and therefore a timeseries comparing snow cover from our experiments with the IMS product would not provide substantial new information beyond that which is already presented in Figure 8, Supplemental Figure S4 and the text.

11) Figure 8: Does it make more sense to have the y axes on these plots be the percentage of sites within each elevation band so it is easier to compare against 25%, 50%, etc of sites?

*Thanks for the suggestion. We will change the y-axis to 'Cumulative fraction of sites with snow disappearance' instead of 'Cumulative sites with snow disappearance'.*

12) Lines 307-320: I think this would be better in the methods. It feels odd to introduce a new dataset in the discussion section.

*Agreed, we will introduce the $DA_{S1}$ dataset in a new subsection of the methodology:*

**2.4.4 Comparison to $SD_{S1}$ DA**

*To compare with previous work that assimilates snow depth retrievals from the S1 change detection algorithm ($SD_{S1}$; Lievens et al. [2022]), we compared output from our two DA experiments with DA output from De Lannoy et al. [2024] (experiment $DA_{S1}$). This $DA_{S1}$ experiment utilized the same DA setup as in $DA_{const}$, with a static observation uncertainty ($\sigma_{obs} = 0.3$ m), but assimilates $SD_{S1}$ retrievals instead of $SD_{ML}$. Here, we utilized 4548 manual SWE measurements collected within the Po River basin (the study domain of De Lannoy et al. [2024]) to compare SWE MAE between the $DA_{const}$, $DA_{var}$, and $DA_{S1}$ experiments.*

**References**

Gabriëlle J. M. De Lannoy, Michel Bechtold, Louise Busschaert, Zdenko Heyvaert, Sara Modanesi, Devon Dunmire, Hans Lievens, Augusto Getirana, and Christian Massari. Contributions of Irrigation Modeling, Soil Moisture and Snow Data Assimilation to High-Resolution Water Budget Estimates Over the Po Basin: Progress Towards Digital Replicas. *Journal of Advances in Modeling Earth Systems*, 16(10), 10 2024. ISSN 1942-2466. doi: 10.1029/2024MS004433.

Devon Dunmire, Hans Lievens, Lucas Boeykens, and Gabriëlle J.M. De Lannoy. A machine learning approach for estimating snow depth across the European Alps from Sentinel-1 imagery. *Remote Sensing of Environment*, 314:114369, 12 2024. ISSN 00344257. doi: 10.1016/j.rse.2024.114369.

S. R. Fassnacht, K. S. J. Brown, E. J. Blumberg, J. I. López Moreno, T. P. Covino, M. Kappas, Y. Huang, V. Leone, and A. H. Kashipazha. Distribution of snow depth variability. *Frontiers of Earth Science*, 12(4):683–692, 12 2018. ISSN 2095-0195. doi: 10.1007/s11707-018-0714-z.

Thomas Grünewald and Michael Lehning. Altitudinal dependency of snow amounts in two small alpine catchments: can catchment-wide snow amounts be estimated via single snow or precipitation stations? *Annals of Glaciology*, 52(58):153–158, 9 2011. ISSN 0260-3055. doi: 10.3189/172756411797252248.

Jordan N. Herbert, Mark S. Raleigh, and Eric E. Small. Reanalyzing the spatial representativeness of snow depth at automated monitoring stations using airborne lidar data. *The Cryosphere*, 18(8):3495–3512, 8 2024. ISSN 1994-0424. doi: 10.5194/tc-18-3495-2024.

Hans Lievens, Isis Brangers, Hans-Peter Marshall, Tobias Jonas, Marc Olefs, and Gabriëlle De Lannoy. Sentinel-1 snow depth retrieval at sub-kilometer resolution over the European Alps. *The Cryosphere*, 16(1):159–177, 1 2022. ISSN 1994-0424. doi: 10.5194/tc-16-159-2022.

J. I. López-Moreno, S. R. Fassnacht, S. Beguería, and J. B. P. Latron. Variability of snow depth at the plot scale: implications for mean depth estimation and sampling strategies. *The Cryosphere*, 5(3):617–629, 2011. ISSN 1994-0424. doi: 10.5194/tc-5-617-2011.

Leah Meromy, Noah P. Molotch, Timothy E. Link, Steven R. Fassnacht, and Robert Rice. Subgrid variability of snow water equivalent at operational snow stations in the western USA. *Hydrological Processes*, 27(17):2383–2400, 8 2013. ISSN 0885-6087. doi: 10.1002/hyp.9355.

M. S. Raleigh, J. D. Lundquist, and M. P. Clark. Exploring the impact of forcing error characteristics on physically based snow simulations within a global sensitivity analysis framework. *Hydrology and Earth System Sciences*, 19(7):3153–3179, 7 2015. ISSN 1607-7938. doi: 10.5194/hess-19-3153-2015.

---

## Author Comment (AC3)

**Reviewer 1**

This manuscript explores the assimilation of a machine learning-derived Sentinel-1 snow depth product into a NOAH land surface model using a dynamically varying observation error. While the approach is innovative and addresses an important limitation in current snow data assimilation systems, the demonstrated improvements over the static error method are relatively modest/marginal and not consistent across space or time. Given the added complexity of implementing a dynamic error model, I do not fully agree with the authors' conclusion that this approach provides a clear performance advantage. The manuscript has potential, particularly if it reframes the findings to emphasize that the benefits of dynamic error observation methods are highly dependent on the pattern and variability of observation errors. One has minimal leverage on the other, and a more thorough error characterization is essential for selecting an appropriate DA strategy for the problem at hand.

We thank the reviewer for their thoughtful and helpful comments and we believe that our revised manuscript is much improved as a result. Our responses below, are in red, with manuscript text in dark red and *additions to the manuscript in italics.*

**Major Comments**

**Comment 1 - Result:**
While the authors claim that the Dvar experiment outperforms $DA_{const}$ in terms of snow depth and SWE estimation, the practical improvement is marginal at best. For snow depth, the spatial ACC increases only slightly—from 0.72 ($DA_{const}$) to 0.73 ($DA_{var}$)—a mere 0.01 gain (Figure 3c), which is unlikely to represent a meaningful enhancement in most applications. The temporal ACC comparison (Figure 3d) shows that just 52% of sites improve with $DA_{var}$ while nearly half do not, and 11% of sites degrade by more than 0.02.

The improvements in spatial and temporal ACC seem small because the model-only run already does a good job at representing the spatial and temporal snow depth patterns, as the model parameterizations and forcing have been previously tuned for optimal results [Brangers et al., 2024]. Indeed, the OL experiment boasts an average spatial ACC of 0.71. However, we can see from Figure 3c that the improvements of $DA_{var}$ compared to the OL are more than double those from $DA_{const}$. We will modify the text in L208-213 to highlight that all experiments well-represent spatial and temporal snow depth patterns in response to the above concern:

*"While the OL experiment already does a good job at representing spatial snow depth patterns (spatial ACC = 0.71), Figure 3c highlights that, for most of the snow season, the $DA_{var}$ experiment offers slight improvements in the representation of these spatial patterns.* Averaged across the entire year, the spatial ACC increases from 0.71 for the OL experiment to 0.72 for $DA_{const}$ and to 0.73 for $DA_{var}$. The greatest improvement in spatial ACC for $DA_{var}$ occurs during the early snow season (November), with values exceeding those of the OL and $DA_{const}$

experiments by 0.058 and 0.047, respectively. From December through April, the spatial ACC for $DA_{var}$ remains approximately 0.021 greater than that of the OL experiment. By mid-April, all three experiments exhibit similar performance in capturing spatial snow depth patterns. *Additionally, both $DA_{const}$ and $DA_{var}$ well-capture temporal snow depth patterns, with average temporal ACC values of 0.68 and 0.72, respectively.* The improvement in temporal ACC for $DA_{var}$ from both the OL and $DA_{const}$ is statistically significant (p < 0.01, Fig. 3d). Across the 948 sites evaluated, 491 sites (52%) have an improved temporal ACC in $DA_{var}$ (> +0.02 compared to $DA_{const}$), while only 103 sites (11%) experience a deterioration in temporal ACC (< -0.02 compared to $DA_{const}$)."

Further, while temporal and spatial ACC are useful metrics for understanding how well the OL and DA experiments capture spatial and temporal snow depth patterns, from a water resource perspective, MAE (or RMSE) and bias are more meaningful statistics. For this reason, most of our Results section focuses on these error and bias metrics. We will further address reviewer concerns regarding MAE and bias improvements in response to the reviewer comments below.

Similarly, the MAE reduction from $DA_{const}$ to $DA_{var}$ occurs at only ∼51% of sites, with 12% worsening. These statistics reveal that the advantage of $DA_{var}$ is not robust or generalizable. The pattern is echoed in the SWE evaluation, where only 56% of sites see improvement in MAE under $DA_{var}$ compared to $DA_{const}$. In other words, nearly half the sites experience no benefit or deterioration, which raises questions about the reliability of the variable uncertainty approach. This is consistent with SDD and SCF evaluation (Figures 7 and 8).

First, a 51% improvement vs. a 12% deterioration at sites is convincing considering the representativeness error of in-situ sites in general. Further, while the improvements seem small at individual in-situ sites, these changes may be amplified when applied across the whole European Alps domain. Previous work has demonstrated that small improvements in SD or SWE lead to further improvements in simulated river discharge [Brangers et al., 2024, De Lannoy et al., 2024]. Regarding sites that experience a deterioration: snow depth retrieved from the Sentinel-1 satellite (either through the conceptual snow depth retrieval algorithm of Lievens et al. [2022], or through the ML-based retrieval of Dunmire et al. [2024] ($SD_{ML}$), is not uniformly robust or necessarily generalizable across all time and space. Many papers have previously demonstrated the limitations of these SAR-based snow depth retrievals [Broxton et al., 2024, Hoppinen et al., 2024, Dunmire et al., 2024]. C-band SAR further appears to be sensitive to snow stratigraphy [Brangers et al., 2024], thus the intensity of snow layering can influence the retrieved snow depth at sites with otherwise similar in-situ snow depth. Given these limitations, there will likely always be locations where the assimilation of a SAR-based snow product deteriorates the simulated snow depth. The $DA_{var}$ experiment is generally more influenced by the observational information than $DA_{const}$ and as such, the performance will be deteriorated in locations where the observations are more inaccurate than the OL experiment. Figure 3b in the manuscript demonstrates that this occurs for a minority of sites. While we have accounted for some of the known limitations of the ML-based snow depth retrieval by not assimilating the $SD_{ML}$ product over glaciers or forested terrain, it is impossible to consider every single grid cell where the S1 retrievals may be unreliable. Thus, there remains a relatively small number of sites that experience a deterioration in the $DA_{var}$

[Figure]

Review Figure 1: Snow depth estimates and independent in-situ measurements from an example site. The dark blue shading represents ±1 standard deviation of the OL ensemble snow depth. The magenta dots represent the assimilated $SD_{ML}$ retrievals, with error bars for the assumed observation error standard deviation from the DA$_{const}$ experiment. Assumed observation error standard deviation from DA$_{var}$ is indicated by the blue error bars.

experiment.

The reviewer also points out that many sites experience no benefit in the DA$_{var}$ experiment. This happens in locations where the DA increments are small, either because the assimilated observations are not substantially different from the prior state, or the uncertainty of the observations is much larger than that of the prior state. For example, we can see in Review Figure 1 that observations in December (black box) are very similar to the simulated snow depth. Additionally, throughout the entire timeseries, $\sigma_{obs}$ from both DA$_{const}$ (blue error bars) and DA$_{var}$ (pink error bars) is substantially larger than the modeled forecast ensemble standard deviation at this site (dark blue shading) which leads to minimal changes in either experiment.

A deterioration in some locations resulting from the DA is perhaps in contrast to the assimilation of lidar data, which are generally more accurate observational products and will lead to more robust and generalizable improvements. However, as demonstrated, assimilating S1-based snow depth observations can still provide benefit, especially as these observations are freely available, and offer frequent, global coverage, qualities which no other snow depth products can offer.

It is also surprising that RMSE was not reported, as it is a standard metric in snow modeling and data assimilation evaluations, and helps better in error magnitude in snow depth and SWE.

MAE is also a standard metric in snow modeling and data assimilation evaluation and both RMSE and MAE are error metrics that indicate how far the predictions are from actual values. We chose to report the MAE instead of RMSE because it gives equal weight to all errors, is more interpretable, and is better for understanding the average error magnitude. In the revised manuscript we will additionally include RMSE values in lines 241-245, and in Figure 5.

"Across the 588 in-situ snow depth measurement sites used for evaluation, the corrections applied in $DA_{var}$ result in snow depth estimates that align more closely with in-situ observations (Fig. 3). The OL experiment yields a site-average MAE of 0.244 m, a *RMSE of 0.300 m*, a bias of 0.113 m, and a Pearson correlation coefficient of 0.75. Both the $DA_{const}$ and $DA_{var}$ experiments show improved performance, with site-average MAE values of 0.237 m and 0.215 m, *RMSE values of 0.292 m and 0.268 m*, and biases of 0.106 m and 0.055 m, respectively."

Despite the narrative of statistical significance (e.g., $p < 0.001$), these results suggest that the magnitude of improvement is small, the spatial consistency is weak, and the operational gain may not justify the added complexity. The authors should contextualize these findings more carefully, perhaps by comparing the computational cost or exploring why improvements are minimal across the full domain.

First, the model-only experiments and atmospheric forcing used here have been previously tuned for optimal performance (in Brangers et al. [2024]) and as such the OL experiment performs reasonably well. If this previous tuning had not been applied, the improvements from the data assimilation would be relatively larger. Nonetheless, the improvements that are presented here are generally larger than other work which assimilates SAR-based snow depth retrievals over the Alps. For instance, Brangers et al. [2024] assimilated the $SD_{S1}$ product over the Western European Alps. They demonstrate that SWE MAE decreases from 134 mm in the OL experiment to 121 mm with DA, a 9.7% reduction in MAE (Figure 4c/f in Brangers et al. [2024]). In our work (which includes substantial additional SWE measurement sites), the SWE MAE decreases from 152 mm in $DA_{const}$ to 132 mm in $DA_{var}$, a 13.2% reduction. In fact, this improvement is comparable to reduction in MAE resulting from the DA in general, as SWE MAE reduces from 176 mm to 152 mm in $DA_{const}$ (-13.6%), indicating that the implementation of $DA_{var}$ can double the error reduction from DA.

Despite small improvements to SD or SWE, Brangers et al. [2024] and De Lannoy et al. [2024] demonstrate that these changes further impact river discharge; thus, seemingly marginal improvements can have a large downstream (pun intended) impact in the land surface model. Additionally, these small, localized improvements will be meaningful when an entire basin or mountain range is considered.

We agree that a stronger contextualization of the results is needed within the discussion. We will incorporate the following paragraph into the Discussion section:

"*While $DA_{var}$ improves performance at most snow depth and SWE measurement sites, some locations see little benefit, or even a deterioration in performance (approximately 12% of snow depth sites and 20% of SWE sites). These degradations are more likely to occur where the*

*$SD_{ML}$ product is less accurate than the OL experiment. To account for known limitations of SAR-based snow depth retrievals, we did not assimilate the $SD_{ML}$ product over dense forests or glaciers, and after March 31. Nevertheless, $SD_{ML}$ remains inaccurate in some places, leading to localized deteriorations when these observations are assimilated. Locations with minimal differences between $DA_{const}$ and $DA_{var}$ typically occur where the observations already agree well with the OL, or where $\sigma_{obs} >> \sigma_f$, thus the DA increments are small, and the model receives limited benefit from the observational information. Despite these spatial inconsistencies, $DA_{var}$ nearly doubles the improvement in absolute SWE error compared to $DA_{const}$. For instance, the SWE MAE decreases from 152 mm in $DA_{const}$ to 132 mm in $DA_{var}$ (-13.2%), while the overall impact of $DA_{const}$ relative to the OL is a 13.6% reduction (176 mm in the OL to 152 mm in $DA_{const}$). Importantly, previous studies have demonstrated that even modest improvements in snow depth or SWE from DA propagate to further improvements in streamflow [Brangers et al., 2024, De Lannoy et al., 2024].*

*...*

*Finally, in $DA_{var}$, Equation 2 ($\sigma_{obs} = m * SD_{ML}$, $m = 0.3$) is used to adapt the standard deviation of the observation error in space and time based on the assimilated snow depth. This relationship is a first-order approximation that assumes that the observation error increases linearly with the observation magnitude; however, $\sigma_{obs}$ could be defined to vary in more complex ways. Future work could explore applying relationships where $\sigma_{obs}$ varies nonlinearly with the assimilated snow depth observation, or statistical parameterizations of $\sigma_{obs}$ depending on other conditions such as elevation, or forest cover. Furthermore, $\sigma_{obs}$ could be directly linked to the $SD_{ML}$ retrieval quality which could be obtained e.g. through error propagation. The effectiveness of a dynamic observation error also depends on the magnitude of the forecast error, as the Kalman gain matrix, which determines the strength of the corrections, depends on both forecast and observation error. To maximize benefits, the observation error, whether static or dynamic, should be properly tuned in relation to forecast error. While most operational systems do not currently include options to dynamically vary the observation error, this functionality is not complicated to incorporate, and the snow-specific MuSA (Multiple Snow Data Assimilation System) system does already provide an option for a user-defined observation error that can vary dynamically [Alonso-González et al., 2022]."*

**Comment 2 - Discussion:**
The discussion attempts to cover many important aspects of the study. However, it overemphasizes statistical significance while underplaying the marginal and spatially inconsistent nature of the improvements. Standard metrics like RMSE are missing to give a clear picture.

Please see our responses to the above concerns for a discussion on the nature of the improvements, an explanation of how we will better contextualize the results within the discussion, and for our inclusion of RMSE metrics.

Key ideas like bias treatment and dynamic observation error comparison with $SD_{S1}$ are introduced without prior mention or clear connection to the results or existing literature.

We discuss bias because it is a known issue in EnKF DA approaches, and thus the concerns

of a biased DA system are important to address here. We will move this discussion into a new a subsection of the Discussion that focuses on these limitations of a bias-blind DA system:

*4.1 Limitations of bias-blind DA systems*

In response to a suggestion from Reviewer 2, we will introduce the comparison with the $\text{DA}_{\text{S1}}$ experiment in the Materials and Methodology section:

*2.4.4 Comparison to $SD_{S1}$ DA*
*To compare with previous work that assimilates snow depth retrievals from the S1 change detection algorithm ($SD_{S1}$; Lievens et al. [2022]), we compared output from our two DA experiments with DA output from De Lannoy et al. [2024] (experiment $\text{DA}_{\text{S1}}$). This $\text{DA}_{\text{S1}}$ experiment utilized the same DA setup as in $\text{DA}_{\text{const}}$, with a static observation uncertainty ($\sigma_{obs}$ = 0.3 m), but assimilates $SD_{S1}$ retrievals instead of $SD_{ML}$. Here, we utilized 4548 manual SWE measurements collected within the Po River basin (the study domain of De Lannoy et al. [2024]) to compare SWE MAE between the $\text{DA}_{\text{const}}$, $\text{DA}_{\text{var}}$, and $\text{DA}_{\text{S1}}$ experiments.*

**Minor Comments**

**Comment 1 - Introduction:**
Since this paper focuses on evaluating different data assimilation (DA) approaches, it would strengthen the manuscript to include a more thorough overview of existing DA algorithm literature, for context and completeness. I recommend adding this near Lines 44–51, where the background on DA methods is introduced.

Agreed. We will provide a more thorough overview of existing DA algorithm literature in this paragraph as follows:

*"One method for assimilating observations into a physical model is via direct insertion, whereby the model's state variables are directly replaced with observations without any statistical blending or error weighting [Rodell and Houser, 2004, Toure et al., 2018]. Increasing in sophistication, optimal interpolation methods, which consider model and observational uncertainty to blend the model and observations using statistically optimal weights [Liston and Hiemstra, 2008], are commonly used at operational centers [Helmert et al., 2018]. Also common among operational centers [Helmert et al., 2018], and one of the most-used DA techniques within the land surface modeling community is the Ensemble Kalman Filter (EnKF; Reichle et al. [2002]). With an EnKF, the background-error covariance is not explicitly computed, but instead estimated using an ensemble of model trajectories. While this ensemble approach is advantageous for high-dimensional, nonlinear systems where an exact computation of the background-error covariance is impractical, the assumption of unbiased, normally distributed model-state errors is often violated for cumulative state variables like snow depth. Despite its reliance on Gaussian assumptions, the EnKF has been extensively used in previous snow data assimilation work [Slater and Clark, 2006, Durand and Margulis, 2006, De Lannoy et al., 2012, Huang et al., 2017, Pflug et al., 2024]. An alternative solution that is commonly used in snow DA, particle batch filters and smoothers are capable of handling non-Gaussian noise*

[Figure]

Review Figure 2: *(Supplemental Figure 1 of revised manuscript) Actual observation error per bin of assimilated snow depth ($SD_{ML}$). The error is computed using 588 independent in-situ measurement sites.*

*and complex posterior distributions. In particular, particle batch smoothers have been commonly applied to create snow reconstructions [Margulis et al., 2015, Baldo and Margulis, 2018, Girotto et al., 2024]."*

**Comment 1 - Methods/Results:**
Justify the use of 0.05 to 1.05 bounds and increasing error with time. It would be helpful to include spatial and temporal figures (e.g., from representative sites) comparing the input observations against independent validation data. This could illustrate the spatial and temporal variability of observation errors and help justify the bounds selected for dynamic error modeling (0.05 to 1.05). For instance, do errors vary systematically with snow depth, such as being lower in shallow snow and higher in deeper snow? As noted by Alonso-González et al. (2024), no data assimilation algorithm is universally superior; performance depends on the data and task at hand. Thus, this figure can further support the choice of the Ensemble Kalman Filter (EnKF) and observation error technique.

To justify the use of an observation error that varies with snow depth we will include Review Figure 2 in the supplementary material, demonstrating how the $SD_{ML}$ error varies with snow depth at our 588 independent measurement sites:

We will also modify the following text to provide this justification, and to justify of our 0.05 and 1.05 lower and upper threshold on the observation error

"... We defined $m$ experimentally by selecting the optimal value when comparing modeled snow depth with in-situ observations in a subset region (6-8 °E, 45-46 °N). Here, we used $m = 0.3$. Equation 2 assumes that $\sigma_{obs}$ varies linearly as a function of assimilated snow depth. Supplemental Figure 1 demonstrates that this assumption is valid at independent in-

[Figure]

Review Figure 3: *(Supplemental Figure 2 of revised manuscript) Standard deviation of the forecast error ($\sigma_f$) per bin of forecast snow depth ($SD_f$). $\sigma_f$ is computed as the standard deviation of the ensembles for the OL.*

*situ measurement sites. For $SD_{ML}$ below 0.25 m, the average error of the $SD_{ML}$ product compared to in-situ measurements is 0.05 (Supplemental Figure 1), and as such we chose this as a minimum threshold value for $\sigma_{obs}$ (Equation 2). Setting this minimum threshold also avoids issues when $SD_{ML}(i,t) = 0$ m. We can see from Supplemental Figure 1 that there are no assimilated snow depths above 3 m at these in-situ measurement sites, making it difficult to characterize the observation error for deeper assimilated snow depths. As such, we also defined an upper threshold for $\sigma_{obs}$ of 1.05 m, corresponding to an assimilated snow depth of 3.5 m (Equation 2). This value was also chosen as an upper threshold because we observed that $\sigma_f$, which represents the uncertainty in the model-only (OL) simulated snow depth, given by the standard deviation of the model ensembles, levels off above 3.5 m snow depth (Supplemental Figure 2). We chose to reflect this feature of the forecast error in our characterization of the observation error."*

**Line to Line Comments**

1) Lines 17–29 would be more effective as a single cohesive paragraph. The current break into two paragraphs disrupts the logical flow and makes the message harder to follow.

We will merge these two paragraphs into a single paragraph.

2) Lines 30-43 discuss various SWE estimation methods, but some other approaches such as spaceborne laser altimetry (e.g., ICESat/ICESat-2), Sentinel-2, and MODIS, are missing and should be acknowledged for completeness. Additionally, the paragraph uses SWE and snow depth somewhat interchangeably (line 42). It would strengthen the clarity to note

explicitly that snow density is required to convert snow depth into SWE.

We will update the text in this paragraph to mention other satellite-based SWE estimation methods:

"Snow depth has also been retrieved using satellite observations, which have the benefit of providing frequent, global coverage [Lievens et al., 2019]. *One approach estimates snow depth by comparing digital elevation models (DEMs) from snow-on and snow-off conditions. These DEMs can be generated from satellite laser altimetry such as ICESat-2 [Enderlin et al., 2022, Deschamps-Berger et al., 2023, Besso et al., 2024] or from very-high-resolution stereoscopic satellite imagery via photogrammetric methods [Marti et al., 2016, Shaw et al., 2020, Deschamps-Berger et al., 2020]. Globally, passive microwave and synthetic aperture radar (SAR) observations are more commonly used to estimate snow depth.* [Kelly et al., 2019, Luojus et al., 2021, Lievens et al., 2022]. However, passive microwave imagery has a coarse spatial resolution..."

We will also clarify the distinction between snow depth and SWE at the beginning of the paragraph:

"Despite the importance of snow within Earth's climate and as a natural resource, accurately quantifying snow mass (or snow water equivalent, SWE) in mountainous, complex terrain remains a challenge. *Because SWE is difficult and costly to directly quantify [Dozier et al., 2016], measurements and retrieval algorithms more commonly focus on snow depth, which is related to SWE via the snow density.*"

3) Line 58-59: particle batch filters and smoothers. can be more computationally expensive given the large number of particles required." While this is generally true for particle filters, it is not necessarily the case for particle batch smoothers (PBS). For example, Alonso-González et al. (2024) showed that PBS was less computationally expensive than EnKF and other particle-based methods across multiple particle counts (100, 200, 300). The authors should either revise the statement to reflect this variability or cite relevant studies to justify the statement.

We will modify the paragraph that contains lines 58-59 in accordance with this reviewer's first minor comment. Our modifications will provide more background details for other various DA methods used in the snow community. In our modification, we will also remove the comment that particle batch filters and smoothers can be more computationally expensive given the number of particles required.

4) Line 67/86: Report the accuracy metrics from Lievens et al. compared to those from Dunmire et al. (2024). Additionally, please clarify that higher accuracy was achieved when evaluated over the European Alps only, as this distinction is crucial for understanding the geographic limitations of these methods.

Following L67, we will add the following sentence to provide metrics for the conceptual [Lievens et al., 2022] and ML-based [Dunmire et al., 2024] models: "*For instance, compared*

*to 798 Alps-wide in-situ measurement sites, the ML model has an average site mean absolute error (MAE) of 0.18 m and an average site bias of -8 mm, compared to an MAE of 0.22 m and a bias of -99 mm for the conceptual model, respectively.*" We will also clarify that that these metrics are valid over the European Alps specifically.

Following L86, we will add the following text to provide metrics for the superiority of ERA5 forcing on modeled snow depth accuracy and bias: "*From Brangers et al. [2024], the ERA5, MERRA-2, and M2CORR atmospheric forcing led to average modeled snow depth MAEs of 0.367 m, 0.404 m, and 0.434 m, and average snow depth biases of -0.07 m, +0.138 m, and -0.363 m, respectively, compared to in-situ measurement stations in the Western European Alps (Figure 10, Brangers et al. [2024]).*"

5) Line 62-73: Clarify that the primary goal of this work is to assess the utility of incorporating dynamic observation errors versus static ones, because there are studies that have compared the performance of different algorithms already.

We will clarify that the primary goal of this work is to assess the utility of incorporating dynamic observation error versus static ones.

6) Line 101-102: specify the reported accuracy metrics (same as 67 and 86). What does better mean?

We will modify the sentence in L102 as follows to include accuracy metrics: "When compared to in-situ snow depth stations and airborne photogrammetry snow depth maps, $SD_{ML}$ is shown to reduce MAE and improve bias compared to $SD_{S1}$ *(MAE reduction from 0.22 m for $SD_{S1}$ to 0.18 m for $SD_{ML}$, bias improvement from -99 mm for $SD_{S1}$ to -8 mm for $SD_{ML}$).*"

7) Line 173-184: The evaluation section introduces Snow Disappearance Date (SDD) and Snow Cover Fraction (SCF) without prior mention or justification in the earlier sections of the manuscript. To improve clarity and coherence, it would be helpful to introduce these variables earlier in the manuscript, explain their relevance to the study objectives.

Snow Disappearance Date and Snow Cover Fraction are relevant to the study objectives in that they are used to **evaluate the DA experiments**. Therefore, we believe that it is most appropriate to introduce these concepts in the Evaluation subsection of Materials and Methodology. We have restructured this section (2.4 Evaluation) to better introduce these concepts and the materials used for this evaluation by adding further subsections. This revised structure will be as follows:

**2.4 Evaluation**
*For each of our three experiments (OL, DA$_{const}$, DA$_{var}$), we utilized a variety of in-situ and satellite-based products to evaluate 1) snow depth, 2) SWE, and 3) snow cover fraction (SCF) and snow disappearance date (SDD).*

**2.4.1 Snow depth evaluation**

...

**2.4.2 SWE evaluation**

...

**2.4.3 SCF and SDD evaluation**

8) Figure 2:

- Why were these two sites chosen? State reason (representativeness, topography, etc) either in the figure description or the result section.

  We will add the following text to the figure caption: "These two sites were chosen because a continuous time series of in-situ data was available, and these sites are generally representative of locations where the DA removes and adds snow."

- Axis can be shared for better readability (general comment for all figures).

  We will share and simplify the x-axis of the graph for better readability (see Review Figure 4)

Line 240-243: State the bias-corrected numbers for the constant as well and compare them to the variable. "Both $DA_{const}$ and $DA_{var}$ also substantially improve the Pearson correlation coefficient..." report numbers.

We will incorporate in-text numbers as suggested for the $DA_{const}$ bias and the Pearson correlation coefficients:

"As a result, the overall average SWE bias decreases from +81 mm in the OL to +18 mm in $DA_{var}$. This bias reduction is significantly greater than that for $DA_{const}$ *(+76 mm bias)*, which only marginally corrects the high bias for low observed SWE, due to minimal model adjustments for shallow assimilated snow depths (e.g., Fig. 5a). Both $DA_{const}$ and $DA_{var}$ also substantially improve the Pearson correlation coefficient *(R = 0.60 for OL, R = 0.72 for* $DA_{const}$*, R = 0.71 for* $DA_{var}$*)*, indicating a stronger correlation with measured SWE."

9) Line 325: The claim that dynamic observation error estimation "had not yet been explored" overlooks prior work that has implemented such approaches (e.g., Alonso-González et al., 2022). While these studies may not compare dynamic and constant errors directly, they do demonstrate prior use of dynamic error treatment in snow science. I suggest rephrasing to acknowledge existing efforts and clarify this study's specific novelty.

We assume that the Alonso-González et al (2022) reference here is referring to the MuSA paper in Geoscientific Model Development. According to this manuscript, a dynamic observation error was not implemented. In Section 2.3, the authors state: "A temporally and spatially static constant scalar corresponding to the assumed observation error variance must

[Figure]

Review Figure 4: (Figure 2 in the Manuscript) Snow depth estimates and independent in-situ measurements at two example sites. (a) Snow depth from $DA_{const}$ (red, left) and $DA_{var}$ (orange, right) compared with the OL (navy) from a measurement station in Austria (13.6228 °E, 47.0944 °N, 1050 m elevation). The shading represents ±1 standard deviation in the model ensembles. The sage green dots represent the assimilated $SD_{ML}$ retrievals, with error bars for the assumed observation error standard deviation ($\sigma_{obs}$, Equation 2). (b) Same as (a), but for a different measurement station in Switzerland (7.7836 °E, 45.9872 °N, 2948 m elevation). *These two sites were chosen due to a lack of gaps in the in-situ measurements and their general representativeness of locations where the DA removes and adds snow.*

be provided for each type of observation that is to be assimilated." [Alonso-González et al., 2022]. In fact, options for a dynamic observation error in MuSA were not incorporated until to v2.1, which was released on May 8, 2024.

However, there does exist other prior work that has utilized dynamic observation errors and as such we will modify this statement accordingly: "While the specification of observation uncertainty substantially influences DA performance, in snow DA systems, this uncertainty is often prescribed as a constant value [Helmert et al., 2018]. *Some previous studies have incorporated dynamic observations errors (e.g., Magnusson et al. [2017], Oberrauch et al. [2024]); however, the utility of dynamic observation errors, relative to an assumed static observation error, in snow DA has not yet been explored prior to this work.*"

**References**

Esteban Alonso-González, Kristoffer Aalstad, Mohamed Wassim Baba, Jesús Revuelto, Juan Ignacio López-Moreno, Joel Fiddes, Richard Essery, and Simon Gascoin. The Multiple Snow Data Assimilation System (MuSA v1.0). *Geoscientific Model Development*, 15 (24):9127–9155, 12 2022. ISSN 1991-9603. doi: 10.5194/gmd-15-9127-2022.

Elisabeth Baldo and Steven A. Margulis. Assessment of a multiresolution snow reanalysis framework: a multidecadal reanalysis case over the upper Yampa River basin, Colorado. *Hydrology and Earth System Sciences*, 22(7):3575–3587, 7 2018. ISSN 1607-7938. doi: 10.5194/hess-22-3575-2018.

Hannah Besso, David Shean, and Jessica D. Lundquist. Mountain snow depth retrievals from customized processing of ICESat-2 satellite laser altimetry. *Remote Sensing of Environment*, 300:113843, 1 2024. ISSN 00344257. doi: 10.1016/j.rse.2023.113843.

I. Brangers, H. Lievens, A. Getirana, and G. J. M. De Lannoy. Sentinel-1 Snow Depth Assimilation to Improve River Discharge Estimates in the Western European Alps. *Water Resources Research*, 60(11), 11 2024. ISSN 0043-1397. doi: 10.1029/2023WR035019.

Patrick Broxton, Mohammad Reza Ehsani, and Ali Behrangi. Improving Mountain Snowpack Estimation Using Machine Learning With Sentinel-1, the Airborne Snow Observatory, and University of Arizona Snowpack Data. *Earth and Space Science*, 11(3), 3 2024. ISSN 2333-5084. doi: 10.1029/2023EA002964.

Gabriëlle J. M. De Lannoy, Rolf H. Reichle, Kristi R. Arsenault, Paul R. Houser, Sujay Kumar, Niko E. C. Verhoest, and Valentijn R. N. Pauwels. Multiscale assimilation of Advanced Microwave Scanning Radiometer–EOS snow water equivalent and Moderate Resolution Imaging Spectroradiometer snow cover fraction observations in northern Colorado. *Water Resources Research*, 48(1), 1 2012. ISSN 0043-1397. doi: 10.1029/2011WR010588.

Gabriëlle J. M. De Lannoy, Michel Bechtold, Louise Busschaert, Zdenko Heyvaert, Sara Modanesi, Devon Dunmire, Hans Lievens, Augusto Getirana, and Christian Massari. Contributions of Irrigation Modeling, Soil Moisture and Snow Data Assimilation to High-Resolution Water Budget Estimates Over the Po Basin: Progress Towards Digital Replicas.

*Journal of Advances in Modeling Earth Systems*, 16(10), 10 2024. ISSN 1942-2466. doi: 10.1029/2024MS004433.

César Deschamps-Berger, Simon Gascoin, Etienne Berthier, Jeffrey Deems, Ethan Gutmann, Amaury Dehecq, David Shean, and Marie Dumont. Snow depth mapping from stereo satellite imagery in mountainous terrain: evaluation using airborne laser-scanning data. *The Cryosphere*, 14(9):2925–2940, 9 2020. ISSN 1994-0424. doi: 10.5194/tc-14-2925-2020.

César Deschamps-Berger, Simon Gascoin, David Shean, Hannah Besso, Ambroise Guiot, and Juan Ignacio López-Moreno. Evaluation of snow depth retrievals from ICESat-2 using airborne laser-scanning data. *The Cryosphere*, 17(7):2779–2792, 7 2023. ISSN 1994-0424. doi: 10.5194/tc-17-2779-2023.

Jeff Dozier, Edward H. Bair, and Robert E. Davis. Estimating the spatial distribution of snow water equivalent in the world's mountains. *WIREs Water*, 3(3):461–474, 5 2016. ISSN 2049-1948. doi: 10.1002/wat2.1140.

Devon Dunmire, Hans Lievens, Lucas Boeykens, and Gabriëlle J.M. De Lannoy. A machine learning approach for estimating snow depth across the European Alps from Sentinel-1 imagery. *Remote Sensing of Environment*, 314:114369, 12 2024. ISSN 00344257. doi: 10.1016/j.rse.2024.114369.

Michael Durand and Steven A. Margulis. Feasibility Test of Multifrequency Radiometric Data Assimilation to Estimate Snow Water Equivalent. *Journal of Hydrometeorology*, 7(3): 443–457, 6 2006. ISSN 1525-7541. doi: 10.1175/JHM502.1.

Ellyn M. Enderlin, Colten M. Elkin, Madeline Gendreau, H.P. Marshall, Shad O'Neel, Christopher McNeil, Caitlyn Florentine, and Louis Sass. Uncertainty of ICESat-2 ATL06- and ATL08-derived snow depths for glacierized and vegetated mountain regions. *Remote Sensing of Environment*, 283:113307, 12 2022. ISSN 00344257. doi: 10.1016/j.rse.2022.113307.

Manuela Girotto, Giuseppe Formetta, Shima Azimi, Claire Bachand, Marianne Cowherd, Gabrielle De Lannoy, Hans Lievens, Sara Modanesi, Mark S. Raleigh, Riccardo Rigon, and Christian Massari. Identifying snowfall elevation patterns by assimilating satellite-based snow depth retrievals. *Science of The Total Environment*, 906:167312, 1 2024. ISSN 00489697. doi: 10.1016/j.scitotenv.2023.167312.

Jürgen Helmert, Aynur Şensoy Şorman, Rodolfo Alvarado Montero, Carlo De Michele, Patricia de Rosnay, Marie Dumont, David Finger, Martin Lange, Ghislain Picard, Vera Potopová, Samantha Pullen, Dagrun Vikhamar-Schuler, and Ali Arslan. Review of Snow Data Assimilation Methods for Hydrological, Land Surface, Meteorological and Climate Models: Results from a COST HarmoSnow Survey. *Geosciences*, 8(12):489, 12 2018. ISSN 2076-3263. doi: 10.3390/geosciences8120489.

Zachary Hoppinen, Ross T. Palomaki, George Brencher, Devon Dunmire, Eric Gagliano, Adrian Marziliano, Jack Tarricone, and Hans-Peter Marshall. Evaluating snow depth retrievals from Sentinel-1 volume scattering over NASA SnowEx sites. *The Cryosphere*, 18 (11):5407–5430, 11 2024. ISSN 1994-0424. doi: 10.5194/tc-18-5407-2024.

Chengcheng Huang, Andrew J. Newman, Martyn P. Clark, Andrew W. Wood, and Xiaogu Zheng. Evaluation of snow data assimilation using the ensemble Kalman filter for seasonal streamflow prediction in the western United States. *Hydrology and Earth System Sciences*, 21(1):635–650, 1 2017. ISSN 1607-7938. doi: 10.5194/hess-21-635-2017.

Richard Kelly, Qinghuan Li, and Nastaran Saberi. 'The AMSR2 Satellite-Based Microwave Snow Algorithm (SMSA): A New Algorithm for Estimating Global Snow Accumulation. In *IGARSS 2019 - 2019 IEEE International Geoscience and Remote Sensing Symposium*, pages 5606–5609. IEEE, 7 2019. ISBN 978-1-5386-9154-0. doi: 10.1109/IGARSS.2019.8898525.

Hans Lievens, Matthias Demuzere, Hans-Peter Marshall, Rolf H. Reichle, Ludovic Brucker, Isis Brangers, Patricia de Rosnay, Marie Dumont, Manuela Girotto, Walter W. Immerzeel, Tobias Jonas, Edward J. Kim, Inka Koch, Christoph Marty, Tuomo Saloranta, Johannes Schöber, and Gabrielle J. M. De Lannoy. Snow depth variability in the Northern Hemisphere mountains observed from space. *Nature Communications*, 10(1):4629, 10 2019. ISSN 2041-1723. doi: 10.1038/s41467-019-12566-y.

Hans Lievens, Isis Brangers, Hans-Peter Marshall, Tobias Jonas, Marc Olefs, and Gabriëlle De Lannoy. Sentinel-1 snow depth retrieval at sub-kilometer resolution over the European Alps. *The Cryosphere*, 16(1):159–177, 1 2022. ISSN 1994-0424. doi: 10.5194/tc-16-159-2022.

Glen E. Liston and Christopher A. Hiemstra. A Simple Data Assimilation System for Complex Snow Distributions (SnowAssim). *Journal of Hydrometeorology*, 9(5):989–1004, 10 2008. ISSN 1525-7541. doi: 10.1175/2008JHM871.1.

Kari Luojus, Jouni Pulliainen, Matias Takala, Juha Lemmetyinen, Colleen Mortimer, Chris Derksen, Lawrence Mudryk, Mikko Moisander, Mwaba Hiltunen, Tuomo Smolander, Jaakko Ikonen, Juval Cohen, Miia Salminen, Johannes Norberg, Katriina Veijola, and Pinja Venäläinen. GlobSnow v3.0 Northern Hemisphere snow water equivalent dataset. *Scientific Data*, 8(1):163, 7 2021. ISSN 2052-4463. doi: 10.1038/s41597-021-00939-2.

Jan Magnusson, Adam Winstral, Andreas S. Stordal, Richard Essery, and Tobias Jonas. Improving physically based snow simulations by assimilating snow depths using the particle filter. *Water Resources Research*, 53(2):1125–1143, 2 2017. ISSN 0043-1397. doi: 10.1002/2016WR019092.

Steven A. Margulis, Manuela Girotto, Gonzalo Cortés, and Michael Durand. A Particle Batch Smoother Approach to Snow Water Equivalent Estimation. *Journal of Hydrometeorology*, 16(4):1752–1772, 8 2015. ISSN 1525-755X. doi: 10.1175/JHM-D-14-0177.1.

R. Marti, S. Gascoin, E. Berthier, M. de Pinel, T. Houet, and D. Laffly. Mapping snow depth in open alpine terrain from stereo satellite imagery. *The Cryosphere*, 10(4):1361–1380, 7 2016. ISSN 1994-0424. doi: 10.5194/tc-10-1361-2016.

Moritz Oberrauch, Bertrand Cluzet, Jan Magnusson, and Tobias Jonas. Improving Fully Distributed Snowpack Simulations by Mapping Perturbations of Meteorological Forcings Inferred From Particle Filter Assimilation of Snow Monitoring Data. *Water Resources Research*, 60(12), 12 2024. ISSN 0043-1397. doi: 10.1029/2023WR036994.

Justin M. Pflug, Melissa L. Wrzesien, Sujay V. Kumar, Eunsang Cho, Kristi R. Arsenault, Paul R. Houser, and Carrie M. Vuyovich. Extending the utility of space-borne snow water equivalent observations over vegetated areas with data assimilation. *Hydrology and Earth System Sciences*, 28(3):631–648, 2 2024. ISSN 1607-7938. doi: 10.5194/hess-28-631-2024.

Rolf H. Reichle, Dennis B. McLaughlin, and Dara Entekhabi. Hydrologic Data Assimilation with the Ensemble Kalman Filter. *Monthly Weather Review*, 130(1):103–114, 1 2002. ISSN 0027-0644. doi: 10.1175/1520-0493(2002)130¡0103:HDAWTE¿2.0.CO;2.

M. Rodell and P. R. Houser. Updating a Land Surface Model with MODIS-Derived Snow Cover. *Journal of Hydrometeorology*, 5(6):1064–1075, 12 2004. ISSN 1525-7541. doi: 10.1175/JHM-395.1.

Thomas E. Shaw, Simon Gascoin, Pablo A. Mendoza, Francesca Pellicciotti, and James McPhee. Snow Depth Patterns in a High Mountain Andean Catchment from Satellite Optical Tristereoscopic Remote Sensing. *Water Resources Research*, 56(2), 2 2020. ISSN 0043-1397. doi: 10.1029/2019WR024880.

Andrew G. Slater and Martyn P. Clark. Snow Data Assimilation via an Ensemble Kalman Filter. *Journal of Hydrometeorology*, 7(3):478–493, 6 2006. ISSN 1525-7541. doi: 10.1175/JHM505.1.

Ally M. Toure, Rolf H. Reichle, Barton A. Forman, Augusto Getirana, and Gabrielle J. M. De Lannoy. Assimilation of MODIS Snow Cover Fraction Observations into the NASA Catchment Land Surface Model. *Remote Sensing*, 10(2):316, 2 2018. ISSN 2072-4292. doi: 10.3390/rs10020316.

---

## Referee Report (RR1)

**General Comment:**

The work has improved from the last draft; however, several conclusions in this paper overstate the magnitude of improvement, terminology drifts into subjective language, and a few structural/citation issues need attention.

**Major Comments:**

I don't agree with the overall conclusion of the paper that DAvar provides substantial improvement over DAcons. The claimed performance gains of DAvar over DAconst are statistically significant but small in absolute terms (<10 cm and, in places, ~2 mm), and not uniform across sites. Please change the framing in the Result/Discussion/Conclusions to emphasize the limited magnitude and spatial inconsistency and to discuss whether the added complexity of DAvar is justified by these gains. Some results are currently summarized with site averages that can be skewed by a few poor sites; compare medians for DAconst vs. DAvar and, if feasible, repeat significance testing on medians. Avoid subjective terms such as "substantial" where differences are on the order of millimeters; replace with exact values. The statement in discussion line 435 that modest snow/SWE improvements translate to streamflow gains is out of context here; withdraw or support it with streamflow evidence. Abstract can also be one or two lines with clear results. Right now, the abstract is vague without stating a clear outcome of the study.

Specific clarifications. Where a "15 mm improvement" is cited (Lines 301–306), specify the metric (likely bias) and state the exact value and sign. Replace "standard WY 2016/2017" with an unambiguous convention (e.g., "Water Year 2017") to avoid seasonality confusion. Across Lines 295–314, report concrete numbers rather than qualitative characterizations. In the concluding sections (Lines 385–391), explicitly acknowledges that DAvar's advantage over DAconst is slight and not pervasive, and suggests that method choice should depend on study goals and acceptable complexity.

**Line-by-line:**

Line 73: add PBS downscaling citations (Bachand 2025; https://doi.org/10.1175/JHM-D-24-0131.1).

Lines 75, 100, 114: insert https://doi.org/10.5194/egusphere-2025-978 wherever relevant to substantiate recent usage and performance.

Section 2.1: Section 2.1 name is misleading: the title ("Noah-MP land surface") suggests a model description, but the text mixes model and forcing details. Rename to "Model setup and data" or similar, and keep model vs. forcing clearly separated

Line 260: compare medians (and re-test significance on medians if earlier tests used means).

Lines 295–300: replace "substantial" with exact mm values and note that benefits must be weighed against DAvar complexity.

Lines 301–314: report actual numbers; avoid "substantial."

---

## Author Response (AR2)

**General Comment:**

The work has improved from the last draft; however, several conclusions in this paper overstate the magnitude of improvement, terminology drifts into subjective language, and a few structural/citation issues need attention.

*We thank the reviewer for their further comments and commitment to improving this manuscript. Our responses below, are in red, with manuscript text in dark red and additions to the manuscript in italics.*

**Major Comments:**

I don't agree with the overall conclusion of the paper that DAvar provides substantial improvement over DAconst. The claimed performance gains of DAvar over DAconst are statistically significant but small in absolute terms (<10 cm and, in places, ~2 mm), and not uniform across sites. Please change the framing in the Result/Discussion/Conclusions to emphasize the limited magnitude and spatial inconsistency and to discuss whether the added complexity of DAvar is justified by these gains.

*We will modify the results section to be as quantitative and objective as possible, removing language such as "substantial", and adding quantification where necessary. We will clarify that improvements are "small, but significant" in places where we test statistical significance. We will further add a few sentences on spatial and temporal inconsistencies in the first paragraph of the discussion:*

"The snow depth estimated from this ML model has been shown to possess superior accuracy compared to prior S1 snow depth retrieval work by Lievens et al. [2022] ($SD_{S1}$) [Dunmire et al., 2024], which has previously been assimilated into the Noah-MP land surface model using an Ensemble Kalman Filter [De Lannoy et al., 2024, Brangers et al., 2024]. *Recent work by Mirza et al. [2025] has questioned the utility of assimilating S1 snow depth retrievals, highlighting inconsistencies in temporal and spatial errors of the $SD_{S1}$ in the Western United States, where less regular S1 data are available. Despite advancements made by $SD_{ML}$, the quality of the ML-based observations assimilated in this study also varies across space and time, which can lead to localized degradations in DA performance (e.g., Fig. 3). Although improving mountain snow depth estimation is an active area of research, progress is limited by the current suite of satellite sensors, which are not specifically designed for snow depth or SWE retrieval. Future DA efforts that incorporate more reliable snow depth or SWE products should reduce these spatial and temporal inconsistencies, improving overall DA performance.*"

*Furthermore, in the previous iteration of review, we added the following text to discuss spatial inconsistencies. We are grateful for the reviewer's previous comments and believe this discussion has improved the manuscript. However, we believe that the current level of discussion sufficiently addresses spatial inconsistencies.*

"While $DA_{var}$ improves performance at most snow depth and SWE measurement sites, some locations see little benefit, or even a deterioration in performance (approximately 12% of snow depth sites and 20% of SWE sites). These degradations are more likely to occur where the $SD_{ML}$ product is less accurate than the OL experiment, *and the $DA_{var}$ experiment more strongly corrects to these inaccurate observations.* To account for known limitations of SAR-based snow depth retrievals, we did not assimilate the $SD_{ML}$ product over dense forests or glaciers, and after March 31. Nevertheless, $SD_{ML}$ remains inaccurate in some places, leading to localized deterioration when these observations are assimilated. Locations with minimal differences between $DA_{const}$ and $DA_{var}$ typically occur where the observations already agree well with the OL, or where $\sigma_{obs} >> \sigma_f$, thus the DA increments are small, and the model receives limited benefit from the observational information."

We will also add the following text to the conclusions to acknowledge the small and spatially inconsistent improvements in $DA_{var}$:

"*However, given limitations of the assimilated satellite-based snow depth product, improvements from the DA, or from the specific implementation of a dynamic observation error in* $DA_{var}$, *are limited in magnitude and not spatially consistent.* As most snow DA work and operational snow DA systems assume that the observational uncertainty is constant in space and time, this work highlights the impact of these assumptions, and the importance of observation uncertainty considerations when designing a DA system. *Future studies should put effort into the consideration of observation uncertainties and the parameterization of observation uncertainty should depend on study goals, the DA system used, and specific characteristics of the assimilated observations.*"

Finally, regarding whether the added complexity of $DA_{var}$ is justified by these gains: very recent work by Gichamo et al. [2025] demonstrates improvements of similar, and even slightly smaller magnitude (SD MAE reduction of 12 mm) when switching from an Optimal Interpolation to a EnKF for snow data assimilation (DA) in NOAA's NWP system, the Global Forecast System (GFS). The authors conclude that "the results are encouraging and motivate implementing ensemble methods for the snow data assimilation in NWP systems, in place of the current OI-based systems". This implementation of ensemble systems would be a much larger step in increased complexity than going from $DA_{var}$ to $DA_{const}$ in our work (which resulted in a SD MAE reduction of 22 mm). In light of recent work highlighting the value of improvements of this magnitude in snow DA systems, we do not believe that achieving comparable, in fact slightly larger, gains should be downplayed here, especially given the minimal added complexity of $DA_{var}$.

Some results are currently summarized with site averages that can be skewed by a few poor sites; compare medians for DAconst vs. DAvar and, if feasible, repeat significance testing on medians.

It is true that the distribution of SD and SWE MAE is right-skewed. As such, we have used a Mann-Whitney U test (an alternative to the two-sample independent t-test when the data is not normally distributed) to test for significantly different distributions in the evaluation metrics. We will specify our use of the Mann-Whitney U test in the results section. Where

relevant, we will also report median MAE values and include significance tests on the medians. Below are the instance where these significance tests will be included:

- For snow depth: "While improvement in MAE from the OL experiment is not significant for $DA_{const}$ (*Mann-Whitney U test p-value = 0.59, median-test p-value = 0.68*), the MAE improvement is *small, but* significant for $DA_{var}$ (*Mann-Whitney U test p-value = 0.001, median-test p-value = 0.03*)".

- For temporal ACC: "The improvement in temporal ACC for $DA_{var}$ from both the OL and $DA_{const}$ is statistically significant (*p < 0.01 for both a Mann-Whitney U test and median-test*, Fig. 3d)."

- For SWE: "Compared with 8,211 manual SWE measurements from 231 different measurement sites across the Alps, the $DA_{var}$ experiment also *offers small, but significant* improvements for SWE MAE compared to both the OL and $DA_{const}$ experiments (*p<<0.001 for both a Mann-Whitney U test and median-test*)."

Avoid subjective terms such as "substantial" where differences are on the order of millimeters; replace with exact values.

We will remove subjective language such as "substantial" and quantify where necessary.

The statement in discussion line 435 that modest snow/SWE improvements translate to streamflow gains is out of context here; withdraw or support it with streamflow evidence.

We will remove this sentence from the discussion.

Abstract can also be one or two lines with clear results. Right now, the abstract is vague without stating a clear outcome of the study.

We will add the following sentences to the abstract to summarize the results:

"*The* $DA_{var}$ *experiment offers small, but significant improvements to snow depth and snow water equivalent (SWE) mean absolute errors (MAE), and slightly reduces snow cover, thereby better matching satellite-based snow cover observations. Compared to an open loop (no DA) experiment (OL), and an experiment with an assumed static observation error (*$DA_{const}$*),* $DA_{var}$ *reduces SWE MAE by 25% and 13%, respectively, compared with over 8000 manual SWE measurements.* This work demonstrates the benefits of machine learning based snow depth retrievals and the impact of incorporating dynamic observation errors in EnKF-based snow DA."

**Specific clarifications.**

Where a "15 mm improvement" is cited (Lines 301–306), specify the metric (likely bias) and state the exact value and sign.

We will specify that the 15 mm improvement refers to SWE MAE.

Replace "standard WY 2016/2017" with an unambiguous convention (e.g., "Water Year 2017") to avoid seasonality confusion.

We will replace "2016/2017" with *"Water Year 2017"*. We have also replaced "2017/2018" (L299) with *"Water Year 2018"* and "2020/2021" (L302) with *"Water year 2021"*.

Across Lines 295–314, report concrete numbers rather than qualitative characterizations.

We will add a quantitative assessment to Lines 295-314. This section will be modified to the text below:

"Across all experiments, SWE typically peaks during the first week of March (March 1–7). Water Year 2017 recorded the lowest modeled SWE in our OL experiment, and correspondingly saw the largest SWE increases in $DA_{var}$ prior to early March, particularly in the Central Alps and Austrian Alps (Fig. 6a). However, $DA_{var}$ SWE improvements were mixed during this year. Of the 41 manual measurements taken between March 1 and March 7, 2017, only 24% demonstrated improved SWE MAE of more than 15 mm in $DA_{var}$. While the DA led to more accurately estimated SWE at some sites (e.g., Supplemental Fig. S3b,d), it resulted in an overestimation of SWE at others (e.g., Supplemental Fig. S3c,e,f). *For example, three measurement sites in Italy (dark pink dots in Fig. 6a) experienced an average increase of 101 mm in added SWE in $DA_{var}$ relative to the OL. The average SWE MAE at these sites increased by 134 mm in $DA_{var}$, indicating that the assimilated $SD_{ML}$ observations overestimate snow at these locations. The degradation is even larger in $DA_{const}$, where the SWE MAE increases by 193 mm compared to the OL. This stronger deterioration arises from the lower assumed $\sigma_{obs}$ in $DA_{const}$ at these locations, which leads to stronger corrections toward the observations. A time series of modeled and observed SWE at one of these sites is shown in Supplemental Fig. S3e.*

The largest SWE reductions from the OL to the $DA_{var}$ experiment occurred during Water Year 2018, particularly in the Bavarian Alps, Swiss Alps, and French Alps (Fig. 6b). In general, the reduced SWE in $DA_{var}$ aligns more closely with in-situ observations (e.g., Supplemental Fig. S4). *The average SWE MAE for in-situ measurements taken between March 1-7, 2018 decreases from 164 mm in the OL, to 137 mm in $DA_{const}$ and 116 mm in $DA_{var}$. In $DA_{var}$, SWE MAE is improved by more than 15 mm in 59% of the 68 manual measurements taken between March 1 and March 7, 2018.*

Water Year 2021 also experienced a large SWE reduction between the OL and $DA_{var}$ experiments, especially in the Swiss Alps and Eastern Dolomites. In the Dolomites region, *where SWE reductions are often greater than 100 mm*, a lack of in-situ observations makes it difficult to assess whether these reductions are realistic. However, limited measurement sites along the Italy-Austria border suggest that the SWE reductions may be too strong (e.g.,

Supplemental Fig. S5d). *For instance, two in-situ measurements sites along the Italy-Austria border (indicated with yellow circles in Supplementary Fig. S5a) have an average SWE decrease of 142 mm in* $\text{DA}_{\text{var}}$, *and a corresponding degradation in SWE MAE of +113 mm. Meanwhile, southwest of these locations, 8 measurement sites in Italy (black box in Supplementary Figure S5a) demonstrate contrasting improvements in* $\text{DA}_{\text{var}}$ *SWE MAE. At these eight sites, SWE decreases by an average of 100 mm in* $\text{DA}_{\text{var}}$, *with a corresponding 74 mm reduction in SWE MAE. This result highlights some of the spatial inconsistencies of the DA improvements, which are likely due to spatial and temporal variation in the quality of the assimilated observations.*

In the concluding sections (Lines 385–391), explicitly acknowledges that DAvar's advantage over DAconst is slight and not pervasive, and suggests that method choice should depend on study goals and acceptable complexity.

In the conclusion, we will add the following text to acknowledge the limited improvement and spatial inconsistencies. We will also suggest that the design of future studies should depend on study goals, the DA system used, and characteristics of the assimilated observations.

"'*However, given limitations of the assimilated satellite-based snow depth product, improvements from the DA, or from the implementation of a dynamic observation error in* $\text{DA}_{\text{var}}$, *are limited in magnitude and not spatially consistent.* As most snow DA work and operational snow DA systems assume that the observational uncertainty is constant in space and time, this work highlights the impact of a these assumptions, and the importance of observation uncertainty considerations when designing a DA system. *Future studies should put effort into the consideration of observation uncertainties and the parameterization of observation uncertainty should depend on study goals, the DA system used, and specific characteristics of the assimilated observations.*"

**Line-by-line:**

Line 73: add PBS downscaling citations (Bachand 2025; https://doi.org/10.1175/JHM-D-24-0131.1).

We will modify the sentence in L73 and add the suggested citation: "In particular, particle batch smoothers have been commonly applied to create snow reconstructions [Margulis et al., 2015, Baldo and Margulis, 2018] *or to downscale model variables such as precipitation [Girotto et al., 2024, Bachand et al., 2025].*"

Lines 75, 100, 114: insert https://doi.org/10.5194/egusphere-2025-978 wherever relevant to substantiate recent usage and performance.

We will add the above citation to the following sentences:

- "Recent studies have used both particle batch smoothers and the EnKF to assimilate

SAR-based snow depth retrievals from Sentinel-1 (S1), thereby improving modeled snow depth, SWE and streamflow compared to in-situ measurements [De Lannoy et al., 2024, Brangers et al., 2024, Girotto et al., 2024, Mirza et al., 2025]."

- "Recent work by Mirza et al. [2025] has questioned the utility of assimilating S1 snow depth retrievals, highlighting inconsistencies in temporal and spatial errors of the $SD_{S1}$ in the Western United States."

- "ERA5 has previously been used as atmospheric forcing in other snow DA studies [Pflug et al., 2024, De Lannoy et al., 2024, Mirza et al., 2025]..."

Section 2.1: Section 2.1 name is misleading: the title ("Noah-MP land surface") suggests a model description, but the text mixes model and forcing details. Rename to "Model setup and data" or similar, and keep model vs. forcing clearly separated

As suggested, we will rename Section 2.1 *"Model setup and data"*. To clearly separate model and forcing we will further add two subsections: *"2.1.1 Noah-MP land surface model"* and *"2.1.2 Atmospheric forcing for Noah-MP"*.

Line 260: compare medians (and re-test significance on medians if earlier tests used means).

We will report the median SD MAE values and test significance on these medians. As described above, we also use a Mann-Whitney U test to test for significantly different distributions of non-normal data.

"Both the $DA_{const}$ and $DA_{var}$ experiments improve these metrics, with site-average MAE values of 0.237 m and 0.215 m (*median values of 0.207 m and 0.185 m*), RMSE values of 0.292 m and 0.268 m, and biases of 0.106 m and 0.055 m, respectively."

"While improvement in MAE from the OL experiment is not significant for $DA_{const}$ (*Mann-Whitney U test p-value = 0.59, median-test p-value = 0.68*), the MAE improvement is slight, but significant for $DA_{var}$ (*Mann-Whitney U test p-value = 0.001, median-test p-value = 0.03*)."

Lines 295–300: replace "substantial" with exact mm values and note that benefits must be weighed against DAvar complexity.

We will modify this text with: "In the OL experiment, we observe a positive bias for low observed SWE and a negative bias for high observed SWE (Fig. **??**c), similar to the bias patterns seen for snow depth. The $DA_{var}$ experiment reduces both biases, with the largest improvements occurring for low observed SWE values. *For instance, for in-situ SWE below 200 mm, the bias is reduced by 52% in* $DA_{var}$ *compared to the OL (OL bias = +166 mm,* $DA_{var}$ *bias = +80 mm), meanwhile the bias in-situ SWE measurements above 600 mm is reduced by 7% in* $DA_{var}$ *(OL bias = -362 mm,* $DA_{var}$ *bias = -335 mm)."*

Lines 301–314: report actual numbers; avoid "substantial."

As discussed above, we will modify to text in these lines to include a more quantitative analysis, avoiding subjective language.

**References**

Claire L. Bachand, Lauren C. Andrews, Tasnuva Rouf, and Manuela Girotto. The Utility of Satellite Snow Depth Observations for Downscaling Hydrologic Variables over the Indus Basin Mountain Ranges. *Journal of Hydrometeorology*, 26(5):555–575, 5 2025. ISSN 1525-755X. doi: 10.1175/JHM-D-24-0131.1.

Elisabeth Baldo and Steven A. Margulis. Assessment of a multiresolution snow reanalysis framework: a multidecadal reanalysis case over the upper Yampa River basin, Colorado. *Hydrology and Earth System Sciences*, 22(7):3575–3587, 7 2018. ISSN 1607-7938. doi: 10.5194/hess-22-3575-2018.

I. Brangers, H. Lievens, A. Getirana, and G. J. M. De Lannoy. Sentinel-1 Snow Depth Assimilation to Improve River Discharge Estimates in the Western European Alps. *Water Resources Research*, 60(11), 11 2024. ISSN 0043-1397. doi: 10.1029/2023WR035019.

Gabriëlle J. M. De Lannoy, Michel Bechtold, Louise Busschaert, Zdenko Heyvaert, Sara Modanesi, Devon Dunmire, Hans Lievens, Augusto Getirana, and Christian Massari. Contributions of Irrigation Modeling, Soil Moisture and Snow Data Assimilation to High-Resolution Water Budget Estimates Over the Po Basin: Progress Towards Digital Replicas. *Journal of Advances in Modeling Earth Systems*, 16(10), 10 2024. ISSN 1942-2466. doi: 10.1029/2024MS004433.

Devon Dunmire, Hans Lievens, Lucas Boeykens, and Gabriëlle J.M. De Lannoy. A machine learning approach for estimating snow depth across the European Alps from Sentinel-1 imagery. *Remote Sensing of Environment*, 314:114369, 12 2024. ISSN 00344257. doi: 10.1016/j.rse.2024.114369.

Tseganeh Z. Gichamo, Clara S. Draper, and Michael Barlage. Improving NOAA's global NWP snow data assimilation by updating to an Ensemble Kalman Filter. *Journal of Hydrology*, 660:133301, 10 2025. ISSN 00221694. doi: 10.1016/j.jhydrol.2025.133301.

Manuela Girotto, Giuseppe Formetta, Shima Azimi, Claire Bachand, Marianne Cowherd, Gabrielle De Lannoy, Hans Lievens, Sara Modanesi, Mark S. Raleigh, Riccardo Rigon, and Christian Massari. Identifying snowfall elevation patterns by assimilating satellite-based snow depth retrievals. *Science of The Total Environment*, 906:167312, 1 2024. ISSN 00489697. doi: 10.1016/j.scitotenv.2023.167312.

Hans Lievens, Isis Brangers, Hans-Peter Marshall, Tobias Jonas, Marc Olefs, and Gabriëlle De Lannoy. Sentinel-1 snow depth retrieval at sub-kilometer resolution over the European Alps. *The Cryosphere*, 16(1):159–177, 1 2022. ISSN 1994-0424. doi: 10.5194/tc-16-159-2022.

Steven A. Margulis, Manuela Girotto, Gonzalo Cortés, and Michael Durand. A Particle Batch Smoother Approach to Snow Water Equivalent Estimation. *Journal of Hydrometeorology*, 16(4):1752–1772, 8 2015. ISSN 1525-755X. doi: 10.1175/JHM-D-14-0177.1.

Bareera N. Mirza, Eric E. Small, and Mark S. Raleigh. Evaluating the Utility of Sentinel-1 in a Data Assimilation System for Estimating Snow Depth in a Mountainous Basin. *Cryosphere Discussions*, 3 2025. doi: 10.5194/egusphere-2025-978.

Justin M. Pflug, Melissa L. Wrzesien, Sujay V. Kumar, Eunsang Cho, Kristi R. Arsenault, Paul R. Houser, and Carrie M. Vuyovich. Extending the utility of space-borne snow water equivalent observations over vegetated areas with data assimilation. *Hydrology and Earth System Sciences*, 28(3):631–648, 2 2024. ISSN 1607-7938. doi: 10.5194/hess-28-631-2024.